# Repetitive Transcranial Magnetic Stimulation Reduces Depressive-like Behaviors, Modifies Dendritic Plasticity, and Generates Global Epigenetic Changes in the Frontal Cortex and Hippocampus in a Rodent Model of Chronic Stress

**DOI:** 10.3390/cells12162062

**Published:** 2023-08-14

**Authors:** David Meneses-San Juan, Mónica Lamas, Gerardo Bernabé Ramírez-Rodríguez

**Affiliations:** 1National Institute of Psychiatry “Ramón de la Fuente Muñiz”, Mexico City 14370, Mexico; meneses.sj.d@imp.edu; 2Center of Research and Advanced Studies of the National Polytechnic Institute, Mexico City 07360, Mexico; mlamas@cinvestav.mx

**Keywords:** depression, epigenetic, dendritic spines, repetitive transcranial magnetic stimulation, dentate gyrus, hippocampus, frontal cortex

## Abstract

Depression is the most common affective disorder worldwide, accounting for 4.4% of the global population, a figure that could increase in the coming decades. In depression, there exists a reduction in the availability of dendritic spines in the frontal cortex (FC) and hippocampus (Hp). In addition, histone modification and DNA methylation are also dysregulated epigenetic mechanisms in depression. Repetitive transcranial magnetic stimulation (rTMS) is a technique that is used to treat depression. However, the epigenetic mechanisms of its therapeutic effect are still not known. Therefore, in this study, we evaluated the antidepressant effect of 5 Hz rTMS and examined its effect on dendritic remodeling, immunoreactivity of synapse proteins, histone modification, and DNA methylation in the FC and Hp in a model of chronic mild stress. Our data indicated that stress generated depressive-like behaviors and that rTMS reverses this effect, romotes the formation of dendritic spines, and favors the presynaptic connection in the FC and DG (dentate gyrus), in addition to increasing histone H3 trimethylation and DNA methylation. These results suggest that the antidepressant effect of rTMS is associated with dendritic remodeling, which is probably regulated by epigenetic mechanisms. These data are a first approximation of the impact of rTMS at the epigenetic level in the context of depression. Therefore, it is necessary to analyze in future studies as to which genes are regulated by these mechanisms, and how they are associated with the neuroplastic modifications promoted by rTMS.

## 1. Introduction

Depression is one of the most common mental disorders reported worldwide and is diagnosed twice as often in women as in men [1]. This disorder is characterized by a lack of interest in daily activities, disturbances in appetite and sleep, loss of pleasure (anhedonia), constant sadness, and in the most severe cases, the formation of suicidal thoughts and the consummation of the same. It is estimated that more than 300 million people worldwide suffer from depression. This figure increased by 25% in the last two years due to the recent COVID-19 pandemic and is predicted to be higher in the coming decades [2].

Depression is a multifactorial disorder resulting from the interaction of several biological, environmental, and psychosocial components [3]. Stress is one factor that increases the risk of developing depression [4], and several animal models based on exposure to stress demonstrated depressive-like behaviors [5]. In addition to the effects of stress, genetic components are also important in the predisposition to suffer depression as it presents a heritability of ≅40% [6]. Among the neurobiological alterations that many patients with depression show, there are changes at the structural and functional level of the cortico-limbic system; in particular, hypofunction and volume reduction in the dorsolateral prefrontal cortex and the hippocampus (Hp) [7,8], and a decrease in the level of hippocampal neurogenesis-related events [9]. Moreover, patients with depression showed a loss in the complexity of pyramidal neurons of the Hp and a reduction in the density of dendritic spines [10,11,12].

Dendritic spines are a crucial structural element existing in synapses to modulate neurotransmission [13]. The formation of dendritic spines follows a process that begins with the remodeling the cytoskeleton to form a long and flexible immature spine that moves in search of contact. When this spine encounters an axon, adhesion molecules are synthesized, along with a few synaptic density proteins. Later, a synapse begins to form with the prior expression of receptors, transporters, and pre- and post-synapse proteins, such as synaptophysin and neurogranin, etc. Finally, establishing the new synapse depends on neural activity [14] that will allow a circuit to function more efficiently. Dendritic spines change their morphology according to their degree of maturation. Thus, they can be classified into either the filopodia (filo)-type spines (immature), thin spines (intermediate), or mushroom (mush) spines (mature). The remodeling of the dendritic spines is sensitive to the environment and is regulated by several epigenetic mechanisms, such as histone modification [14,15] and deoxyribonucleic acid (DNA) methylation [16]. These mechanisms are relevant as they are involved in the molecular neurobiology of depression. For instance, in a murine model of social defeat stress, an increase in histone deacetylation in the amygdala and Hp was found, an effect which can be reversed using the histone deacetylase inhibitor (HDAC) MS-275 [17]. In addition, imipramine, an antidepressant drug, increased histone H2 [18], H3 [19], and H5 [20] deacetylation in different models of stress. Moreover, fluoxetine (FLX) increases histone H3 acetylation and decreases depressive-like behaviors [21]. Also, stress decreases the expression of different histone methyltransferases (HMT) in the *nucleus accumbens* [22], and the increase in the HMT G9a induces an antidepressant effect. Thus, the inhibition of this enzyme generated depressive-like behaviors in rodents [22]. At the DNA methylation level, the expression of stress- (NR3C1 [21]) and neuroplasticity- (BDNF [23], SLC6A4 [24]) associated genes are altered under the conditions of depression. In addition, several studies reported global changes in DNA methylation in serum samples from patients with depression [25].

Moreover, due to the complexity of depression, several therapeutic strategies have been used, among which neurostimulation techniques, such as repetitive transcranial magnetic stimulation (rTMS), have stood out in recent decades. rTMS is a noninvasive indirect method that is used to induce changes in nerve excitability, modification in brain connectivity, and for the induction of an antidepressant effect [26]. The frequencies most used to treat depression are those of 10–20 Hz [27]. However, there are clinical data that suggest that 5 Hz protocols impart the same therapeutic effects as those produced with higher frequencies [28,29].

It is known that rTMS alters the brain levels of neurotransmitters, such as glutamate [30], GABA [31], 5-HT, and dopamine [32]. Furthermore, rTMS increases BDNF expression and signaling in the motor cortex [33], promotes neuroplastic changes at the structural level, and promotes the generation of new neurons [34]. In addition, rTMS induces dendritic remodeling in the hippocampus [35], and at the functional level, it favors long-term potentiation in the hippocampus (Hp) and frontal cortex (FC) [36]. However, there is little information available regarding the involvement of epigenetic modifications in the beneficial effects of rTMS. For instance, rTMS at 15 Hz induced the expression of dopaminergic receptors and different synapse proteins in the FC involving histone acetylation in mice [36].

Despite the antidepressant effect of rTMS having being previously demonstrated in clinical and preclinical studies, the mechanisms of action by which this neurostimulation technique causes its antidepressant effect is still unclear, and the evidence available at the epigenetic level is scarce. Thus, this study aimed to explore the effects of rTMS (5 Hz) on dendritic spine remodeling associated with global DNA methylation and histone modifications (histone H2B acetylation and histone H3 trimethylation) in the FC and dentate gyrus (DG) of the Hp in a murine model of depression induced by chronic stress. The 5 Hz rTMS protocol was based on previous protocols of our group, which reported the antidepressant effect of rTMS at this frequency in patients diagnosed with major depression [29,37].

## 2. Materials and Methods

### 2.1. Experimental Design

Female BALB/c mice were exposed to the chronic unpredictable mild stress (CUMS) protocol for eight weeks (with a previous week of acclimatization) to induce depressive-like behaviors to further evaluate the antidepressant effects of rTMS (5 Hz). The application of rTMS and the administration of FLX and the sham treatment were performed from day 35 for 4 weeks. To evaluate the development of depressive-like behaviors during the CUMS protocol, the coat state (CS) of mice was measured weekly, as well as their sucrose consumption using the sucrose preference test (SPT). Twenty-four hours after the end of the CUMS protocol and the therapeutic interventions, the forced-swim test (FST) was performed to confirm the final effects of these treatments on depressive-like behaviors in mice. Two hours after completing the FST, mice were sacrificed by decapitation (as outlined in Figure 1A).

### 2.2. Animals

Eight-week-old female BALB/c strain mice were randomly divided into four groups, which were as follows: control (Ctrl), CUMS+sham, CUMS+rTMS, and CUMS+FLX, respectively. The “n” per group = 7–10. Mice were obtained from the Instituto Nacional de Psiquiatría Ramón de la Fuente Muñiz (INPRFM). All experimental procedures were performed following the NOM-062-ZOO-1999 and were approved by the Ethical Committee of the INPRFM.

### 2.3. Chronic Unpredictable Mild Stress Protocol

The stress protocol used was based on the protocol proposed by Willner et al. in 1992 [38] and modified by Vega-Rivera et al. in 2016 [38]. This protocol consisted of applying one to three stressors per day of one, without patterns, applied in the morning, afternoon, and/or evening, considering that no more than two stressors with a duration of 12–24 h can be applied within the same day (Appendix A). The stressors used and their durations are displayed in Table 1.

### 2.4. Repetitive Transcranial Magnetic Stimulation (rTMS)

A MagPro R30 (MagVenture, Alpharetta, GA, USA) stimulator with a coil adapted for rodents (Cool-40) was used (as shown in Figure 1B). The dimensions of the coil were as follows: width: 52 mm, length: 54 mm, and height: 42 mm, respectively. It has a maximum change in the magnetic field strength per time (dB/dt) of 80 kT/s (5 mm from the coil’s surface) (Figure 1C). Prior to starting the rTMS application, the motor threshold was identified per individual without anesthesia. For this purpose, single pulses were applied to the motor cortex with a magnetic stimulation output (MSO) of 10% of the maximum, followed by subsequent increases of 2% of the maximum until an involuntary and evident motor response was obtained in the mouse limbs. The MSO value that evoked the motor response was considered the motor threshold and was the same that was subsequently applied to generate an antidepressant effect. In the fourth week after starting the CUMS protocol, when the mice showed depressive-like behaviors, the application of rTMS in the left frontal cortex was initiated (Figure 1D). For this purpose, the induction point of the coil was placed above the heads of the mice (≈2 mm behind the mouse eye and put the coil on its head with an inclination of ≈140°) (Figure 1E). The application of rTMS was made without anesthesia by immobilizing the mouse manually during the stimulation session. Mice were previously acclimatized for one week before starting their rTMS application to reduce handling stress and ensure animal immobility. During this period, mice were immobilized and exposed to the coil for 5 min daily. Subsequently, rTMS was applied five days a week for four weeks (20 sessions). Each session consisted of using 30 trains of 5 Hz stimulation. Each train included ten stimuli with 10 s intervals between the stimulation trains with 1500 pulses per day (30 trains × 10 stimuli × 5 Hz) [29,37]. For mice in the sham group, the coil was placed in the same position as the experimental group, with the only difference being that the electromagnetic pulses were not applied.

### 2.5. Drugs

For the positive control group (CUMS+FLX), FLX (Abcam, Cambridge, UK) was administered as an antidepressant treatment (10 mg/kg/day). FLX was dissolved in 0.9% NaCl (LaPisa, Mexico City, México). The administration route was intraperitoneal (i.p) once a day from the fourth week of the CUMS protocol for 28 days [39].

### 2.6. Behavioral Tests

#### 2.6.1. Evaluation of the Coat State

The CS of rodents was recorded weekly during the nine weeks of the experiment (including the acclimation week). The CS of mice was assessed as a measure of the CUMS-induced alterations in their motivation towards self-care behavior. The CS score was based on the scale used by Tanti et al. in 2012 [39], where a value of 0 represents a good condition (smooth and shiny fur, without patches on the coat), 0.5 an intermediate one (slightly fluffy fur with some patches), and 1, a bad condition (fluffy fur on most of the body with slight patches on the skin), respectively. Measurements were made on eight areas of the mouse’s body: the head, neck, upper and lower back, abdomen, tail, and legs, respectively. The final score obtained was the result of the average of these measurements. This index has been pharmacologically validated in previous studies [40].

#### 2.6.2. Sucrose Preference Test

To complement the assessment of the coat state and to obtain another parameter that would serve as an indicator of the development of depressive-like behaviors throughout the CUMS protocol, the SPT was also performed weekly. The SPT consisted of depriving the mice of food and water for 18 h, and then evaluating their consumption of a sucrose solution (1%) or natural water in an individual box (19.05 × 29.21 × 12.7 cm). Before the first sucrose consumption assessment (basal week), mice were exposed to the taste of the sugar solution for one week in their home cage, followed by one week of adaptation to the individual box with the two troughs. The position of the troughs was changed weekly in order to ensure taste preference and to not place preference [41]. The parameters evaluated in this test were the net sucrose consumption and the sucrose preference index, which takes water consumption into account and is calculated as follows: sucrose consumed/(sucrose consumed + natural water consumed) × 100.

#### 2.6.3. Forced-Swim Test

Twenty-four hours after the end of the exposure to stress and the application of the treatments, the FST was performed to evaluate the “despair” or the ability of mice to cope with inescapable stressors [42]. A 30 cm high and 8 cm in diameter glass cylinder was used for this test, which was filled to approximately 15 cm depth with water at 25 ± 1 °C. The test was performed in a single session for 6 min per individual. The total immobility time and the number of immobility episodes were quantified with the help of the behavioral analysis program Any maze version 6.19. The criteria for considering immobility in the mice were that 90% of their whole body remained motionless for a minimum period of 2.5 s.

### 2.7. Brain Processing for Golgi Cox and DNA Isolation

Once the mice were euthanized, their brains were extracted, and 4 of 7 or 10 brains were destined for histological processing. The left hemispheres were used for Golgi Cox impregnation, and the right hemispheres were destined for immunohistochemistry and immunofluorescence, respectively (as displayed in Figure 2). Moreover, 3 brains from each group were used to dissect the DG (Figure 2). Dissection was performed under stereoscopy with the tissue in maintenance solution (sucrose 210 mM, KCl 2.8 mM, MgSO_4_ 2 mM, Na_2_HPO_4_ 1.25 mM, NaHCO_3_ 25 mM, glucose 10 mM, CaCl_2_ 1 mM, and MgCl_2_ 1 mM at 2 °C) during the whole process. Brain regions were dissected to extract DNA using the silica column separation technique. Subsequently, the DNA samples were used to quantify the levels of 5-methyl-cytosine (5mC) with ELISA.

### 2.8. Golgi–Cox Staining

Golgi–Cox staining (Golgi–Cox PK401 kit, NeuroTechnologies, Inc. Columbia, MD, USA) was performed to determine the microstructure of the dendrites in the pyramidal cells of the FC and the granular cells of the DG. Brains were fixed for 4 min with 4% paraformaldehyde and were then washed with 1× phosphate-buffered saline (PBS). Subsequently, the tissue was impregnated with a potassium dichromate/mercury chloride solution for 14 days and then transferred to a potassium chromate solution for 7 days. The brain was then sectioned with a microtome (Leica, Buffalo Grove, IL, USA) to obtain 200 µm-thick slices that were incubated with ammonium hydroxide for 10 min to make the staining develop. Finally, the tissue was dehydrated with sequential dilutions of alcohol (95%, 75%, and 50%, respectively) and Milli-Q water for 5 min for each wash [43].

#### Morphological Analysis of the Dendritic Spines and Dendritic Tree Complexity

To perform the morphological analysis of the dendritic spines and trees, we captured images within the pyramidal neurons in layers II/III of the FC and granular cells of the DG of the Hp, in accordance with the Allen brain atlas [44]. To analyze the density of the dendritic spines, images of 20 secondary dendrites were captured per mouse. The number of spines in a length of 10 µm was quantified with ImageJ version 1.53c. For the morphological classification of the spines observed, the length and width of each spine was measured using the Reconstruct version 1.1 program. Dendritic spines were categorized according to their geometric characteristics into three types of morphologies, which were as follows: (a) filopodia: length > 2 µm (immature spines), (b) thin: length ≤ 1 µm and a ratio of length/width ≤ 1 (intermediate spines), and (c) mushroom: width > 0.6 µm (mature spines) [45]. Finally, in order to evaluate the complexity of the dendritic trees of neurons, pictures of eight cells per mouse were taken, and the number of intersections per dendritic tree was quantified using Sholl analysis (which consisted of 11 circles of 40 µm in diameter from the soma) and with the help of the free Cell Target software (latest version 7.1.0).

### 2.9. Immunohistochemistry

#### 2.9.1. Tissue Processing for Immunohistochemistry

Brains were first fixed in 4% paraformaldehyde for 24 h, and then transferred to a 30% sucrose solution in phosphate-buffered saline (PBS 1×) until sectioning. Brains were sectioned into 40 µm-thick sagittal slices with a microtome (Leica, Buffalo Grove, IL, USA). The slices were stored in a cryoprotectant solution (composed of 25% ethylene glycol, 25% glycerol, and 50% 1× PBS) until the point of their processing with free-floating immunohistochemistry [46].

#### 2.9.2. Antibodies and Immunodetection

Immunohistochemistry was performed for the pre-synapse protein synaptophysin (SYP) and the post-synapse protein neurogranin (RC3) as an indirect measurement of synapse protein expression. For the measurement of the global epigenetic marks, we detected the immunoreactivity of the acetylated histone H2B in lysine 16 (H2BK16ac), trimethylated histone H3 in lysine 9 (H3K9me3), and the modified nitrogenous bases, 5-methylcytosine (5mC) and 5-hydroxy-methylcytosine (5hmC). The primary antibodies used were as follows: anti-synaptophysin (1:1000, Cat: SAB4502906, Sigma-Aldrich, St. Louis, MO, USA), anti-RC3 (1:250, Cat: NBP229349, Novus Biologicals, Littleton, CO, USA), anti-histone H2B acetyl K16 (1:2000, Cat: ab177427, Abcam, Cambridge, UK), anti-Histone H3 trimethyl K9 (1:2000, Cat: ab8898, Abcam, Cambridge, UK), anti-5-methylcytosine, (1:500, Cat: ab10805, Abcam, Cambridge, UK), and anti-5-hydroxymethylcytosine (1:700, Cat: ab106918, Abcam, Cambridge, UK). The secondary antibodies used were biotinylated goat anti-rabbit (SYP, 5-mC, H3K9me3), biotinylated donkey anti-goat (1:250), (RC3, H2BK16ac), and biotinylated goat anti-rat (5hmC) at a dilution of 1:250 (Jackson Immunoresearch, West Grove, PA, USA). The signal was amplified using the peroxidase method. Finally, the brain slices were mounted in a Neo-Mount medium (Cat: 109016. Merck, State of Mexico, Mexico).

#### 2.9.3. Measurement of Immunoreactivity

Brain slices containing the Hp and FC were imaged (using an ICC50 camera coupled to an optical microscope; Leica, Buffalo Grove, IL, USA) with Leica suite application software v. 3.4.0. Immunoreactivity was quantified in arbitrary units (A.U) with ImageJ v. 1.53c software as the areas with the highest number of pixels per area (integrated optical density). All photographs were taken at the same brightness, saturation, and contrast levels. The images were calibrated (image calibration min = 0, max = 2.708) and then analyzed in an 8-bits format. Five squares of 200 × 200 µm per slice were considered along layers II/III to measure the extent of immunoreactivity in the FC. In total, 5–6 slices per mouse were analyzed. The immunoreactivity was then measured in the hippocampus’s molecular layer, with ten squares of 50 × 50 µm per slice being considered, and 5–7 slices analyzed per mouse. The background values per brain section analyzed were calculated in regions lacking immunoreactivity. The data reported are the mean of the immunoreactivity measurements obtained from the squares analyzed for each region.

### 2.10. Immunofluorescence and Image Analysis

The population of mature neurons expressing 5mC in the DG was analyzed via the means of immunofluorescence in arbitrary units with double labeling. The first examined was the neuronal nuclear protein (NeuN) as a marker of mature neurons, and the second observed was 5mC as an indicator of DNA methylation. The primary antibodies used were anti-NeuN (1:1000, Cat: MAB377, Millipore, Mexico City, Mexico) and anti-5-methylcytosine (1:500, Cat: ab214727, Abcam). 4′,6-diamidino-2-phenylindole (DAPI), (1:1000, Cat: ab228549, Abcam) was used to stain the nuclei of the hippocampal dentate gyrus cells.

Images were acquired using a Carl Zeiss LSM900 confocal microscope with a 40× objective using immersion oil (Leica, Wentzler, Germany). Images were acquired in a z-plane with a thickness of 0.5 µm between each slice (thirteen slices). Thirty DAPI/NeuN/5mC-positive cells per dentate gyrus were analyzed (3 hippocampi per mouse, totaling 90 cells per individual). Analysis was performed on MERGE imaging in the orthogonal plane at maximum projection intensity using Zeiss ZEN Blue version 3.5 software [47]. The mean fluorescence intensity and the area in each of the nuclei of the analyzed cells were quantified. Data are reported as the mean of the ratios of the area over the fluorescence intensity per cell.

### 2.11. DNA Extraction and Methylation Analysis

The QIAamp DNA Micro 56304 kit and the protocol proposed by the manufacturer were used for DNA extraction, after which elution DNA was stored at −20 °C. For DNA quantification, an Epoch microplate reader and Take3 (BioTek, Winooski, VT, USA) was used to determine the amount of extracted DNA. The 5mC colorimetric assay was performed with the DNA samples using the kit “Methylated DNA Quantification colorimetric” (Cat: ab117128, Abcam). The absorbance of the reaction was measured at 450 nm with a multidetector (GloMax Discover, Promega, CA, USA).

### 2.12. Statistics

Statistical analyses were performed using SigmaStat version 12.3, and graphs were created using GraphPad Prism version 9.3. The results are presented as the mean ± standard error of the mean. To determine the type of distribution (parametric or nonparametric) of our data, we performed the Shapiro–Wilk normality test and the test for the equality of variances. Following this, the differences in means between groups were analyzed with one- or two-way ANOVAs (repeated measures in some cases) or one-way ANOVAs on ranks, based on the nature of the data, followed by the Bonferroni or Student–Newman–Keuls post hoc test, respectively. Differences were considered statistically significant with a *p* ≤ 0.05. To identify the main variables that could explain the effects of the treatments, we performed a principal component analysis (PCA) using GraphPad Prism version 9.3. All continuous variables were included for the PCA. Data were standardized prior to analysis to prevent the introduction of bias due to differences in the measurement scales. The Kaiser criterion was used to determine the number of principal components to be retained, selecting those with eigenvalues greater than 1. The PCA results were interpreted by observing the load patterns on the principal components and were used to identify groups of related variables.

## 3. Results

### 3.1. Repetitive Transcranial Magnetic Stimulation and Fluoxetine Reverse Depressive-like Behaviors Generated by Chronic Unpredictable Mild Stress

The evaluation of the CS revealed that the coat deteriorated in chronically stressed mice compared to the control mice (*p* < 0.001). This observation was uncovered from the second week of CUMS exposure (*p* = 0.009), reaching a plateau in the level of deterioration 4 weeks after the CUMS protocol was initiated. However, mice showed decreased coat deterioration from the first week of treatment with 5 Hz rTMS compared with the sham group (*p* = 0.002), but FLX did not exhibit a significant decrease in the CS (*p* = 0.065). The effect of both interventions on the CS was evident after the second week of treatment (*p* = 0.004; *p* < 0.001), and it remained significantly lower for the rest of the treatment compared with the sham group (*p* < 0.001; *p* < 0.001) (Figure 3A; two-way repeated measures ANOVA. Factor A (treatment: F3,287 = 103, *p* < 0.001), factor B (time: F8,287 = 259, *p* < 0.001), and interaction (treatment x time: F7,287 = 19, *p* < 0.001)). The sucrose preference test revealed similar effects in the mice exposed to the CUMS protocol, showing a decreased sucrose solution consumption level compared to the control mice after the third week of chronic stress (*p* = 0.008); the effects caused by CUMS was sustained for the rest of the experiment (*p* < 0.001). However, in weeks 7 and 8 following the initiation of the CUMS protocol, 5 Hz rTMS and FLX reversed this anhedonic behavior compared with the chronically stressed mice without antidepressant treatment (week 7: *p* = 0.010, *p* = 0.002, respectively; week 8: *p* < 0.001, *p* < 0.001, respectively) (Figure 3B; two-way repeated measures ANOVA. Factor A (treatment: F3,359 = 32, *p* < 0.001), factor B (time: F8,359 = 7, *p* < 0.001), and interaction (treatment x time: F24,359 = 4, *p* < 0.001))). When evaluating the sucrose preference index, a behavior like that of sucrose consumption was observed. A significant decrease in this parameter from weeks 3 to 8 was observed in the sham mice compared to the Ctrl group (weeks 3, 4, 6, and 8: *p* < 0.001. week 5: *p* = 0.013). Meanwhile, this effect was reversed in mice treated with rTMS and FLX compared to the sham group from two weeks after treatment application until the end of the CUMS protocol (rTMS week 6: *p* = 0.012, weeks 7 and 8: *p* < 0.001, respectively. FLX week 6: *p* = 0.009, weeks 7 and 8: *p* < 0.001, respectively). (Figure 3C; two-way repeated measures ANOVA. Factor A (treatment: F3,359 = 27, *p* < 0.001), factor B (time: F8,359 = 8, *p* < 0.001), and interaction (treatment x time: F24,359 = 4, *p* < 0.001).

In addition, the antidepressant-like effect of 5 Hz rTMS was evaluated in the FST by analyzing two parameters: immobility episodes (Figure 3D) and time (Figure 3E). Interestingly, both of these parameters were found to positively correlate (r^2^ = 0.3240, *p* = 0.0002, N = 37), indicating the association of these parameters as indicators of despair-like behavior (Figure 3F). 

Thus, the induction of the CUMS protocol caused a significant increase in the number of episodes of immobility (*p* = 0.021) and a tendency to increase the immobility time (*p* = 0.185) compared with the control group. In both parameters, namely immobility episodes and time, 5 Hz rTMS reversed the increments caused by CUMS (episodes of immobility: *p* = 0.003 and time of immobility: *p* = 0.025, respectively), whereas FLX significantly reversed the increase in the number of bouts of immobility (*p* = 0.022) (Figure 3C; one-way ANOVA, F3,36 = 6, *p* = 0.002), and showed a tendency to decrease the time of immobility (*p* = 0.299) (Figure 3D; one-way ANOVA, F3,36 = 3, *p* = 0.026). These results suggest that both 5 Hz rTMS and FLX reverse the CUMS-induced depressive-like behaviors.

### 3.2. rTMS Modifies the Density of the Dendritic Spines and the Complexity of the Dendritic Trees in the Pyramidal Neurons of the Frontal Cortex and the Granule Cells of the Hippocampal Dentate Gyrus

The mechanisms underlying the antidepressant effects of rTMS and FLX may involve modifications in structural neuroplasticity, such as dendritic remodeling in the brain structures of the limbic system, such as the FC and the Hp. Thus, we analyzed the complexity of the dendritic trees in the pyramidal cells of the FC (Figure 4A,B). Sholl analysis of the FC revealed that the CUMS protocol caused fewer intersections in the pyramidal cells compared to the control non-stressed group (*p* < 0.001). The decrease in the number of intersections was reversed by the treatment with 5 Hz rTMS (*p* < 0.001) or FLX (*p* < 0.001). Moreover, the number of intersections along the dendrite was higher in the control non-stressed group compared to the mice exposed to the CUMS protocol at distances of 60 to 120 μm from the soma (60, 80 μm: *p* < 0.001, 100 μm: *p* = 0.005, and 120 μm: *p* = 0.006, respectively). Similar observations were found in the mice treated with 5 Hz rTMS (40 μm: *p* = 0.008, 60 μm: *p* = 0.002, 80 μm: *p* < 0.001, 100 μm: *p* = 0.005, and 120 μm: *p* = 0.005, respectively) and FLX (40 μm: *p* = 0.023, 60 μm: *p* = 0.016, 80 μm: *p* = 0.002, 100 μm: *p* < 0.001, and 120 μm: *p* = 0.001, respectively) (Two-way ANOVA. Factor A (treatment: F3,191 = 26, *p* < 0.001), factor B (distance from the soma: F11,191 = 123, *p* < 0.001), and interaction (treatment × distance: F33,191 = 1, *p* = 0.34)). 

Regarding the complexity of the dendritic trees of the granular cells in the DG (Figure 4C,D), Sholl analysis revealed similar results to those found in the FC. Mice that were chronically stressed showed fewer intersections compared to the control non-stressed mice (*p* < 0.001). Again, 5 Hz rTMS (*p* < 0.001) and FLX (*p* < 0.001) reversed the decrement in the number of intersections produced by the induction of the CUMS protocol. Furthermore, the number of intersections along the dendrite was found to be higher in the control non-stressed group than in the mice exposed to CUMS at distances from 80 to 180 μm from the soma (80 μm: *p* = 0.003, 100–160 μm: *p* < 0.001, and 180 μm: *p* = 0.004, respectively). Similar observations were found in the mice treated with 5 Hz rTMS (40 μm: *p* = 0.020, 60 μm: *p* = 0.002, 80 μm: *p* = 0.004, 100 μm: *p* = 0.002, 120 μm: *p* = 0.003, 140 μm: *p* = 0.004, and 160 μm: *p* = 0.010, respectively) and FLX (60 μm: *p* = 0.046, 80 μm: *p* = 0.014, 100 μm: *p* < 0.001, 120 μm: *p* = 0.007, 140 μm: *p* = 0.004, and 160 μm: *p* = 0.033, respectively) (Two-way ANOVA. Factor A (treatment: F3,191 = 44, *p* < 0.001), factor B (distance from the soma: F11,191 = 47, *p* < 0.001), and interaction (treatment x distance: F33,191 = 1, *p* = 0.17)). Thus, the number of intersections observed at different distances from the soma indicates that 5 Hz rTMS and FLX reversed the decreased dendritic complexity caused by the exposure to the CUMS protocol. In addition to the quantification of the number of intersections in the dendritic tree, we quantified the density of the dendritic spines (DSs) in the FC (Figure 4E,F) and DG (Figure 4H,I). Mice that were chronically stressed showed a decrease in the density of the DSs in the FC (q = 4.20) and DG (q = 5.92). Both treatments, 5 Hz rTMS and FLX, reversed the effects of CUMS on the DS density in the FC (q = 4.43 and q = 4.89, respectively, one-way ANOVA on ranks. H = 10, df = 3, *p* = 0.020) and in the DG (q = 3.67 and q = 4.43, respectively, one-way ANOVA on ranks. H = 9, df = 3, *p* = 0.034). These results suggest that both 5 Hz rTMS and FLX reverse the decreased density of the DSs in the FC and DG caused by the CUMS protocol. 

Therefore, we analyzed the effects of these treatments on the types of DS present based on their morphology in the FC and DG (Figure 4G,J, respectively). In the FC (Figure 4G), we did not find an interaction between the factors (Figure 4G; two-way ANOVA. Treatment X DS type: F6,47 = 2, *p* = 0.074). However, the main effect of treatment (F3,47 = 10, *p* < 0.001) confirmed that CUMS decreased the number of all types of DS, regardless of their type, compared to the control group (*p* < 0.001). This effect was reversed by 5 Hz rTMS (*p* = 0.013), but FLX tended to reverse the effect of chronic stress (*p* = 0.082). Also, the main effect of the DS type (F2,47 = 88, *p* < 0.001) showed that in all groups, mushroom-like spines were the most abundant compared with filopodia (*p* < 0.001) and thin (*p* < 0.001) DS types (Figure 4G). However, in the DG (Figure 4J), CUMS only affected the thin DSs (*p* < 0.001) compared with the control non-stressed group. Interestingly, 5 Hz rTMS did not reverse the decreased number of thin DSs (*p* = 1.0) but increased the number of filopodia (*p* = 0.004) and mushroom (*p* = 0.005) DSs compared with chronically stressed mice. In the case of mice treated with FLX, we found an increased number of thin (*p* = 0.039) and mushroom (*p* = 0.001) DSs compared with stressed mice (Two-way ANOVA. Factor A (treatment: F3,52 = 13, *p* < 0.001), factor B (DS type: F2,52 = 56, *p* < 0.001), and interaction (treatment x DS type: F6,52 = 5, *p* < 0.001)).

Together, the results of the density and morphology of the DSs suggest that 5 Hz rTMS or FLX reversed the dendritic remodeling alterations produced by the CUMS protocol. The increased number of mushroom DSs suggests that both interventions, 5 Hz rTMS and FLX, favored the process involved in the maturation of the DSs.

### 3.3. rTMS Increased the Immunoreactivity of Synaptophysin without Influencing the Immunoreactivity for Neurogranin

The structural modifications in the dendritic trees and DSs may be accompanied by changes in the expression of synapse proteins, such as SYP and RC3 [48,49]. In this work, the immunoreactivity of these proteins, as an indirect marker of their expression, was evaluated in layer II/III of the FC and the molecular layer (ML) of the DG, as these cellular layers are where the dendritic trees of the pyramidal neurons of the FC and the granule cells of the DG are found [50,51].

RC3 in the FC (Figure 5A,B) and the ML in the DG (Figure 5C,D) did not reveal any significant differences among the groups (FC: one-way on ranks ANOVA, H = 4, df = 3, *p* = 0.210, and ML: one-way ANOVA, F3,15 = 2, *p* = 0.106, respectively). 

In the case of SYP in the FC (Figure 5E,F), it increased in the mice treated with the 5 Hz rTMS (*p* = 0.013) or FLX (*p* < 0.001) compared to the stressed mice. In the case of FLX, this treatment was found to be significantly different compared with the Ctrl (*p* < 0.001) and rTMS (*p* = 0.020) groups (one-way ANOVA, F3,15 = 23, *p* < 0.001).

In the ML in the DG (Figure 5G,H), compared to the sham group, SYP immunoreactivity increased in the rTMS group (*p* = 0.048) and showed a trend towards chance in the FLX group (Figure 5H) (*p* = 0.077; one-way ANOVA, F3,15 = 4, *p* = 0.026). These results suggest that both treatments, 5 Hz rTMS and FLX, favored the increased expression of SYP, a protein located on the presynaptic vesicles [52]. 

### 3.4. Repetitive Transcranial Magnetic Stimulation or Fluoxetine Induce Differential Effects on the Immunoreactivity of the Histone H2B (K16) Acetylation and the Histone H3 Trimethylation

To obtain an insight into the regulations of the behavioral and dendrite structure through epigenetic modifications produced by rTMS and FLX, we evaluated the immunoreactivity of H2BK16ac in the FC and DG (Figure 6A–D). H2BK16ac was assessed as an activating epigenetic mark associated with structural changes in the dendrites. For this purpose, we quantified the immunoreactivity of this epigenetic marker in layer II/III of the FC and the DG of the hippocampus (Figure 6A). In the FC (Figure 6B), FLX was observed to increase H2BK16ac immunoreactivity compared to the control (*p* = 0.032), sham (*p* < 0.001) and rTMS (*p* = 0.012) groups (one-way ANOVA, F3,14 = 12, *p* < 0.001). In the DG (Figure 6C), we did not observe any significant differences for this epigenetic marker among these groups (Figure 6D; one-way ANOVA F3,15 = 3, *p* = 0.068).

In addition, H3K9me3 was evaluated as an epigenetic repressor mark at the histone modification level (Figure 6E). In layer II/III of the FC (Figure 6F), the rTMS group revealed an increased H3K9me3 immunoreactivity compared to the Ctrl and sham groups (*p* = 0.006 and *p* = 0.040, respectively). However, FLX did not significantly modify this mark (*p* = 0.104; one-way ANOVA, F3,14 = 7, *p* = 0.006). In the DG (Figure 6G,H), there were no significant differences observed in the immunoreactivity of H3K9me3 among these groups (*p* = 0.121; one-way ANOVA F3,15 = 2).

These results suggest that chronic stress did not influence H2BK16ac and H3K9me3 in the FC or DG. However, rTMS and FLX induced differential effects in the immunoreactivity of both epigenetic marks.

### 3.5. rTMS Increased DNA Methylation Globally and at the Level of Mature Neurons in the Hippocampal Dentate Gyrus

In addition to exploring the acetylation and methylation processes of the histones, we analyzed DNA methylation, which is part of the main repressor epigenetic marks in the FC and the DG. Then, we quantified the immunoreactivity of 5mC as a methylation indicator (Figure 7) and 5hmC as a demethylation indicator (Figure 8) in the FC and DG of the Hp.

The immunoreactivity of 5mC in the FC (Figure 7A,B) did not reveal any significant differences among the groups (one-way ANOVA, F3,14 = 1, *p* = 0.544). However, in the DG (Figure 7C,D), rTMS increased the immunoreactivity for this marker compared to the control group (*p* = 0.015; one-way ANOVA, F3,14 = 5, *p* = 0.014). To confirm the increased 5mC immunoreactivity in the DG, we quantified the 5mC concentration using ELISA (Figure 7E). CUMS decreased the concentration of 5mC compared to the control group (*p* = 0.028). The effect caused by CUMS exposure was reversed by rTMS (*p* = 0.015), but not by FLX (*p* = 1.00; one-way ANOVA, F3,11 = 15, *p* = 0.001). This result strongly suggests that rTMS specifically increased global DNA methylation in the hippocampus. Although this result talked about the global methylation in the hippocampus, it did not let us know the differences in the levels of 5mC expression in the mature neurons of the DG. These cells showed increased DSs and dendritic tree complexity following rTMS treatment. Thus, we performed triple immunofluorescence staining to identify the neuronal proteins NeuN and 5mC (Figure 7F,G). The 5mC expression in NeuN cells revealed that CUMS significantly decreased DNA methylation compared to the control group (q = 3.683). Still, applying rTMS or FLX reversed this decrement (q = 4.006 and q = 3.703, respectively. One-way ANOVA on ranks, H = 8, df = 3, *p* = 0.050). These data confirm that the rTMS increased global DNA methylation levels in the DG. Interestingly, mature neurons increase DNA methylation. While FLX treatment did not reverse the decrease in global DNA methylation, it did reverse the decrease in the methylation of the mature neurons in the hippocampal DG. This may suggest that rTMS may act on other types of cells residing in the hippocampus, whereas FLX seems to exhibit a more significant effect on the mature neurons of the DG.

### 3.6. Stress-Exposure Increased 5hmC in the Frontal Cortex

DNA methylation is a dynamic process that is reversible through demethylation, which together is a crucial part in the control of gene expression to maintain cell homeostasis [53]. Therefore, demethylation levels were assessed by measuring the immunoreactivity of 5hmC in layer II/III of the FC and the DG of the hippocampus. At the FC (Figure 8A,B), the CUMS-exposed groups significantly increased the immunoreactivity of 5hmC compared with the control (*p* < 0.001). Mice treated with rTMS or FLX did not show significant changes compared to the sham group (*p* = 1.00), (one-way ANOVA, F3,15 = 20, *p* < 0.001), (one-way ANOVA, F3,15 = 20, *p* < 0.001). In the DG (Figure 8C,D), we did not find a significant difference among the groups (one-way ANOVA on ranks, DF = 3, H = 5, *p* = 0.147). These data indicate that DNA demethylation is not affected by rTMS or FLX, and that CUMS increased the immunoreactivity of this epigenetic mark in the FC, but not in the DG.

### 3.7. Principal Component Analysis of the Most Representative Parameters at the Level of Dendritic Remodeling, Synapse Proteins, Histone Modification, and DNA De/Methylation

To integrate the parameters that were assessed previously and reduce the number of variables to identify those that explain most of the effects observed from each experimental group, we performed a PCA. The Kaiser criterion was used to determine the number of principal components to be retained, selecting those with eigenvalues greater than one. Our analysis revealed that the first two components explained more than 50% of the variance of the data (64%). However, component three was also analyzed to identify the patterns of the similarities and differences between the groups more clearly.

The first component (*x*-axis) groups control mice together with those that received antidepressant treatment (rTMS and FLX) and separates the sham group. In comparison to the first component, component two (*y*-axis) explains the differences between the groups who received antidepressant treatment (rTMS and FLX) and the Ctrl group (Figure 9A). On the other hand, component three (axis “*y*” Figure 9B) separates the rTMS group from the rest of the other groups.

Component one explained 41% of the variance of the data; the variables with the highest contribution (54%) were as follows: the number of the mush and filo spines, total spine density, and the number of intersections of the pyramidal neurons in the FC in addition to the total spines and crossings of the granular neurons in the DG (Figure 9C). Component 2 explained 23% of the variance of the data and separates the groups with antidepressant treatments (rTMS and FLX) from the Ctrl group. The variables with the highest contribution (51%) in this component were: the immunoreactivity of 5hmC and SYP in the FC and the immunoreactivity of 5hmC, H2BK16ac, and RC3 in the hippocampus, in addition to the levels of DNA methylation in the mature neurons in the DG (Figure 9C). Component 3 explained 16% of the variance of the data. The variables with the most significant contribution (58%) in component 3 were H3K9me3 in the FC, 5mC levels, and the number of thin spines in the DG (Figure 9C).

Overall, these data may suggest that variables associated with dendritic remodeling mainly explain the effects of stress. In contrast, variables related to epigenetic changes (activation marks) explain the impact of antidepressant treatments, primarily the effects of FLX, while variables associated with repressor epigenetic marks (H3K9me3 and DNA methylation) mainly demonstrate the effects of rTMS.

## 4. Discussion

### 4.1. Findings

In this work, we explored the effects of 5 Hz rTMS under different levels, including the behavioral level, dendritic remodeling, synapse proteins, and at the level of epigenetic mechanisms (histone modification, and DNA methylation and demethylation) in the FC and Hp in female BALB/c mice exposed to the CUMS protocol. Our findings indicate that rTMS at 5 Hz reversed the depressive-like behaviors generated by CUMS. rTMS favored the formation and maturation of DSs in the FC and Hp. Furthermore, rTMS modulated changes in different epigenetic signatures, such as histone H3 trimethylation in the FC, along with an increase in global DNA methylation and mature neurons in the DG of the Hp. In addition, according to the PCA, these epigenetic mechanisms (histone trimethylation and DNA methylation) are the most important variables that could explain the behavioral effects of rTMS.

### 4.2. rTMS at 5 Hz Reverses the Behavioral Alterations Generated by Chronic Stress

Depression is a neuropsychiatric disorder that has been strongly associated with stress since many depressive episodes occur due to different stressors [54]. The CUMS protocol is one of the paradigms that has been most used in the modeling of depressive-like behaviors, as it reproduces the neurochemical, morphological, and neuroplastic alterations that have been reported in patients with depression [55].

In our work, we reported that CUMS generated anhedonia, “despair” behavior, and deterioration in the CS in BALB/c mice. The CS is an indicator of the motivational and self-care behaviors, which display pharmacological validation using different antidepressant drugs [56]. Here, we found a similar coat deterioration reported in male BALB/c and C57BL/6 mice [40]. FLX and rTMS improved self-care behaviors and reversed the deterioration of the coat state caused by CUMS. The positive effect of FLX on the CS of the mice exposed to stress has been previously reported by other authors [40,57,58,59], while the effect of 5 Hz rTMS on the CS was recently shown in female Swiss Webster mice exposed to the CUMS protocol [60]. At the level of anhedonia, the application of rTMS, both at 15 Hz (21,000 pulses per session) [61,62], and 5 Hz [63], increased sucrose consumption in chronically stressed mice.

Moreover, 5 Hz rTMS reversed the immobility induced by CUMS, which is consistent with the findings of a previous study performed in Sprague Dawley rats [63]. Similar results have been produced using higher frequencies, such as 15 Hz [61], 10 Hz [61], and 25 Hz [62] of rTMS, respectively. Together, the evidence described here along with our results suggest that the antidepressant effect of rTMS can produced with a wide range of frequencies, with 5 Hz as the minimum frequency to induce antidepressant-like effects. However, we consider that the significance of our findings lies in demonstrating the therapeutic efficacy of a relatively underexplored rTMS stimulation protocol at the preclinical research level. Specifically, we utilized a protocol involving 300 pulses of 5 Hz administered over 20 sessions. This approach provides valuable insights into the potential of this specific rTMS protocol for the treatment of depression, as was previously probed at the clinical level by our group [29,64].

### 4.3. 5 Hz rTMS Reverses Dendritic Atrophy at the Level of the Dendritic Trees and Spines Generated by Chronic Stress in the Frontal Cortex and Dentate Gyrus of the Hippocampus

Our results suggest that 5 Hz rTMS promotes dendritic remodeling in both the FC and the Hp. These two brain structures exhibit a significant level of neuroplastic sensitivity in response to environmental stimuli, such as stress [65]. Our data concerning the decrease in the density of dendritic spines and the simplification of the dendritic trees generated with the CUMS protocol are consistent with those reported by other authors in analyses performed on neurons from different hippocampal regions, such as the CA3, CA1 [65], and DG regions [12], as well as in the medial prefrontal cortex [66]. In previous studies, it has been shown that stress decreased the amount of mature dendritic spines (mushroom-like) in the rodent hippocampus [12,67]. In contrast to the data found in our work, the thin-type spines were the most affected by stress. This may be because in these previously mentioned studies, their analyses were conducted on another type of hippocampal cell (pyramidal neurons of the CA3 or CA1 region), in addition to the fact that, as pointed out by Qio et al., 2016 [10], data concerning the effect of stress on the morphology of the dendritic spines of the granule cells in the DG varies in response to the nature of the stressors applied and the duration of the protocol. On the other hand, our data obtained from the FC analysis are consistent with what has been previously reported by several authors [10,12], confirming that this structure is one of the most sensitive to stress at the level of dendritic plasticity, specifically in the loss of dendritic spines, and in the atrophy of the dendritic trees of the pyramidal cells of layer II/III [65].

The data available in the literature on the effects of rTMS on dendritic remodeling in the FC are null and in the Hp are diverse. For example, Cambiaghi et al. in 2020, showed similar results to ours, reporting that rTMS application at 1 Hz (120 pulses per session) for five days generated an increase in the lengths of the dendrites of the granule cells in the hippocampal DG, as well as an increase in the spine density and complexity of the dendritic trees [35]. Similarly, it has been demonstrated that rTMS at 1 Hz (300 pulses per session) at 1.14 Teslas favors the growth of dendrites in primary hippocampal cultures. In contrast, rTMS at the same frequency, but with a somewhat higher intensity (1.55 Teslas), decreases this effect [68]. In cell cultures of CA1 neurons, rTMS at 10 Hz (900 pulses per session) increased their excitatory synaptic strength but did not modify the density of the dendritic spines [69]. This difference in the results obtained may be due to the utilization of different rTMS protocols with different experimental models.

Our study contributes novel information regarding the positive impact of rTMS at 5 Hz on dendritic remodeling in neurons located in the frontal cortex (FC) and dentate gyrus of the hippocampus in a mouse model of chronic unpredictable mild stress (CUMS). These findings suggest that the antidepressant effects of rTMS may be mediated not only by neurogenesis, as previously documented by Ueyama et al. (2011) [34] and Czéh et al. (2002) [70], but also by the restructuring of dendrites and their spines.

### 4.4. rTMS May Promote the Increase in the Immunoreactivity of Presynaptic Proteins in the Pyramidal Neurons of the Frontal Cortex and the Granular Cells of the Dentate Gyrus

Synapse formation is a process that requires the expression of several proteins involved in neurotransmission and the maintenance of connections between neurons. The synthesis or degradation of these proteins is key to the generation of different types of plasticity, such as long-term potentiation and depression [71]. In this work, our data suggest that rTMS at 5 Hz promotes presynaptic activity in the pyramidal neurons of the FC and the granular cells of the DG. These changes at the level of SYP may be associated with dendritic remodeling, since it has previously been shown that SYP favors the formation of synapses dependent on neural activity in hippocampal neuron cultures [72] and participates in the development of dendritic trees and in the formation of dendritic spines in the pyramidal neurons of the prefrontal cortex of macaques [73].

Interestingly, the effects of rTMS on synapse markers and neurotrophic factors have been reported across various brain regions and experimental models. For instance, Ma et al. (2014) demonstrated that rTMS at 1 Hz with 400 pulses per session increased the expression and transcription of several synapse markers, including SYP, PSD95, and GAP43, in the CA1 and CA3 regions of the hippocampus in an aging model [74]. This suggests that low-frequency rTMS can promote synaptic plasticity and enhance synaptic connectivity. Additionally, a study by Qian et al. (2023) showed that rTMS at 20 Hz with 800 pulses per session reversed the loss of the synapse marker SYP in a brain injury trauma model, specifically in the motor cortex [75]. Moreover, this stimulation protocol increased the transcription of BDNF, TrkB (the receptor for BDNF), and the glutamatergic NMDA receptor. These findings suggest that high-frequency rTMS can modulate both synaptic markers and neurotrophic factors, potentially promoting synaptic regeneration and functional recovery following brain injury [76]. Under non-pathological conditions, it has also been shown that rTMS (at 0.5 Hz 500 pulses per session) increased the level of synaptic density in the hippocampus and that these changes are mediated by CaMKII activity [77].

In general, the changes in the immunoreactivity of SYP found in this work may indicate that the rTMS at 5 Hz induced synaptic plasticity in the FC and the Hp. According to the location of the presynaptic terminals, it can be speculated that rTMS favors the connectivity of the granule cells with the pyramidal neurons of layer II/III in the FC. The granule cells are typically found in these cortical layers, and their axons can connect with the dendrites of the pyramidal neurons [78,79], and regulate the activity of the pyramidal neurons [80]. In the hippocampus, since the axons connecting to the dendrites of the granular cells of the DG come from the neurons in the entorhinal cortex [81], it can be speculated that rTMS promotes the presynaptic connectivity of the perforant pathway, which is involved in different memory types [82,83], and regulates the hippocampal activity. The increased connectivity of this pathway could be associated with the improvement in the cognitive abilities reported with the use of rTMS in rodents [84] and healthy humans [85], which are functions that have also been found to be altered in patients with depression [86].

### 4.5. FLX Favors Histone Acetylation Whereas 5 Hz rTMS Increases Histone Trimethylation in the Frontal Cortex

Histone modification is a post-translational mechanism that reshapes the structure of chromatin, regulates gene expression, and participates in neuronal development, plasticity, and behavior [87]. In our work, it was found that CUMS tended to decrease H2BK16ac in the FC. Previous studies have suggested that low histone acetylation is associated with the molecular pathophysiology of depression, since it has been found that in rodents exposed to stress by social defeat, there is a decrease in the histone acetylation of the H2 [18], H3 [19], and H5 [20] histones in the hippocampus and the *nucleus accumbens*.

Our study observed that fluoxetine (FLX) can reverse the decrease in histone H2B acetylation and even increase it beyond basal levels. Interestingly, similar findings have been reported for histone H3 acetylation using a model of sodium butyrate-induced depression in C57BL/6J mice [21].

At the level of histone trimethylation, we found that stress does not modify H3K9me3 in the DG. Other authors evaluated this epigenetic mark in stress models, such as Covintong et al. in 2011, who pointed out that social defeat stress decreases histone H3 demethylation and the enzymes in charge of it (G9a) in the *nucleus accumbens* in C57BL/6J mice [22], which is different to what we found in this work. We explained this inconsistency in the results being due to the trimethylation measurements being made in different brain regions and the stress protocols used. At the level of the FC, which is the region where rTMS was applied, we found that this technique increased the levels of this repressor mark, suggesting that the increase in this epigenetic marker may be associated with an antidepressant effect, as indicated by previous studies in which the overexpression of HMT G9a decreased the depressive-like behaviors generated by stress [22]. 

Contrary to this, other studies indicate that stress generated an increase in histone H3 demethylation in the hippocampus, which is associated with the generation of depressive-like behaviors [88], and that the increase in histone H3 demethylation in the *nucleus accumbens* generated an antidepressant effect [89]. These differences with our data may be due to the stress protocol employed and the brain regions analyzed. Acetylation and methylation of histones may be one of the pathways by which rTMS induces dendritic and synaptic plasticity in the FC and Hp. However, histone modification is probably not the only epigenetic mechanism affected by rTMS, so it was also decided to study DNA methylation.

### 4.6. DNA Methylation in the DG Is Favored by rTMS and May Be the Mechanism by Which Dendritic Remodeling Is Promoted in the Mature Neurons

DNA methylation is particularly abundant in the brain, where it plays a crucial role in gene silencing, in the regulation of neuronal activity, and in the facilitation of neuroplasticity, which are essential processes for brain function and development [90].

In this work, we explored the effect of rTMS on global DNA methylation in the FC and the DG. Our data suggest that chronic stress decreases global methylation levels in the DG, which may be associated with the decrease in the methylation of genes involved in the stress response, such as the pro-opio-melanocortin gene [91], vasopressin [92], and the corticotropin-releasing hormone gene [93], as has been reported in other stress models. 

Our data also indicate that rTMS reverses the decrease in global methylation generated by stress. This phenomenon may be associated with the pro-neurogenic effect, the favoring of dendritic maturation, and the improvement in synaptic connectivity reported with this technique, since such processes are neuroplastic changes which are highly dependent on the suppression of various genes, and which allow the differentiation and specialization of neurons in the hippocampus [94]. We found that there is indeed an increase in the DNA methylation levels of the mature DG neurons, in which the analyses of their dendritic trees and spines were performed. These results are consistent with that published by Zocher et al. in 2021, who reported that de novo DNA methylation controls neuronal maturation and dendritic growth in cultured adult hippocampal neurons [47]. These authors pointed out that the process of neuronal differentiation requires genetic and epigenetic molecular programs that allow precursor cells of the dentate gyrus to specialize and become mature neurons. In our case, we found that rTMS may be promoting dendritic spine maturation [34,35,74] through increased DNA methylation. However, it is currently not known which genes are being repressed, thus favoring these neuroplastic changes.

### 4.7. Chronic Stress Induces Demethylation in the Frontal Cortex Independently of rTMS and FLX Application

Demethylation is responsible for the remodeling of the methylome. It serves as a defense mechanism against exogenous DNA expression [95] and functions as an activating mark as it promotes gene expression by inhibiting methylation. This mechanism is essential for cellular reprogramming and for the ability of cells to contend with changes in their environment [96]. In this work, we measured 5hmC levels as a marker of demethylation, and found that chronic stress increased demethylation levels in the FC but not in the DG. It has previously been reported that demethylation may function as a DNA repair mechanism when the methylation process is altered. Concordant with this, Xin, et al. in 2013 suggested that in the postmortem prefrontal cortex of patients with depression, there was a tendency to increase 5-hydroxy-methyl-cytosine levels, although these data were not conclusive. The authors explained that this tendency to increase demethylation may be associated with reduced methylation levels in depressed patients [97]. In addition, it is known that high levels of cytosines and oxidative stress increase demethylation through activation of the GADD45-alpha protein complex [98]. Part of the physiological alterations caused by chronic stress is precisely the induction of an inflammatory state in the brain that elevates the levels of proinflammatory cytokines [99] and of reactive oxygen species, conditions that elevate the risk of developing depression [100], and which could be the mechanisms by which we found increased demethylation in our work. Another important result to note is that neither rTMS nor FLX reverses this effect in our model, indicating that they do not act at the level of demethylation, which is in contrast to rTMS, which does increase methylation. In this regard, it is interesting in that these changes in demethylation are only observed in the FC, where no significant changes were found concerning DNA methylation. This is contrary to what was observed in the DG, where stress decreases methylation while rTMS increases it, but where there are no significant changes observed regarding demethylation. There is a dynamic balance between demethylation and methylation. However, these processes are regulated by different enzymes, meaning that they do not necessarily have to be maintained at opposite levels, as reported in our results. Although the role of DNA methylation has been studied in the neurobiology of different psychiatric disorders [101], information regarding the role of demethylation in depression is scarce, and even more so in the context of rTMS. Therefore, this work provides new information about demethylation in the FC as part of the epigenetic alterations underlying depression, and as a molecular mechanism independent of the antidepressant effect of rTMS.

## 5. Conclusions

Our study showed depressive-like behaviors in female BALB/c mice through exposure to a chronic stress protocol. Within this model, we observed significant changes in dendritic and synaptic plasticity and alterations at the epigenetic level. Interestingly, rTMS and FLX exhibited similar behavioral and neuroplastic effects, indicating their comparable antidepressant efficacy. However, our investigation into the impact of these treatments on epigenetic markers revealed distinct molecular mechanisms, despite achieving the same antidepressant outcome.

Specifically, our data suggest that rTMS predominantly facilitates epigenetic repressor mechanisms, whereas FLX primarily triggers activation mechanisms. Through principal component analysis (PCA), we proposed that these epigenetic changes were associated with the observed modifications in these dendritic and synaptic structures. Our findings suggest that gene silencing, mediated by histone H3 trimethylation in the frontal cortex (FC), and DNA methylation in the mature DG neurons, promotes the structural modifications of the DSs of the FC and hippocampus (Hp), consequently activating the cortico-limbic system. This mechanism may partially explain the antidepressant effect of 5 Hz rTMS. It is important to note that while our study identified modifications in these specific epigenetic marks induced by rTMS, it is likely that these mechanisms, in turn, regulate other processes and gene expression. These orchestrated interactions ultimately contribute to the formation of synapses, which are crucial for overall brain function and plasticity.

## 6. Limitations and Perspectives

One of the limitations of this work is that only changes in histone modification and DNA (de)methylation were explored globally, and the expression levels of genes that these mechanisms could regulate were not measured. Nevertheless, we consider that the information provided by this work is new and valuable as it is a first approximation of the epigenetic effect of rTMS at 5 Hz in an animal model of depressive-like behaviors. The data generated in this work add new pieces to the puzzle of the mechanism of action of the antidepressant effect of rTMS, with which it has been intended that in the future, all the biological effects of this therapeutic alternative will have been elucidated. In this way, it would be possible to improve and expand the use of rTMS in the biomedical area (psychiatric, neurodegenerative, motor diseases, etc.). However, there are still many questions regarding the molecular mechanisms of rTMS in depression, such as which genes are being regulated by histone methylation and trimethylation, whether these global epigenetic changes occur in all cell populations or only in neurons, what other epigenetic mechanisms are involved, and how they interact with each other to regulate neuronal plasticity in the hippocampus and even in different brain areas. All these questions open new possibilities and research that, in the future, may help us to better understand and apply rTMS and other neurostimulation techniques.

## Figures and Tables

**Figure 1 cells-12-02062-f001:**
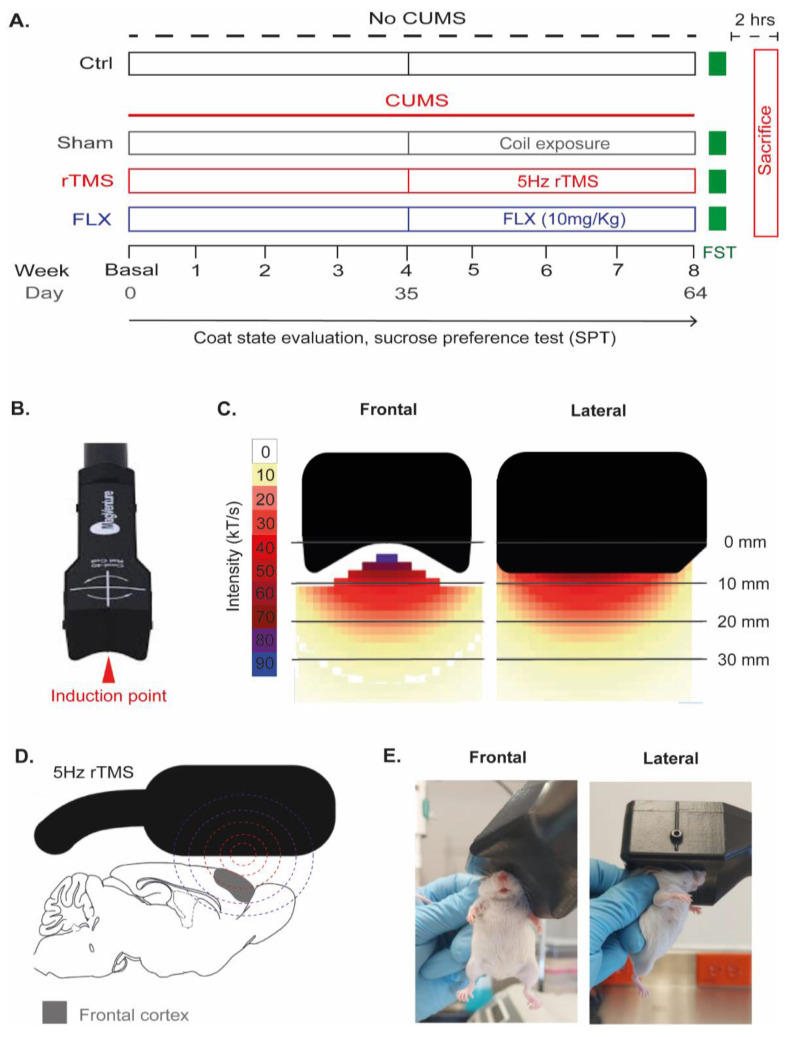
Experimental design and application of repetitive transcranial magnetic stimulation. (**A**) Timeline outlining the duration of the chronic unpredictable mild stress (CUMS) protocol, the moments of the therapeutic intervention applied, the application of the behavioral tests, and the conditions of each experimental group. Experimental groups consisted of the control without stress (Ctrl), sham exposed to chronic stress (CUMS), mice exposed to CUMS treated with 5 Hz rTMS, and mice exposed to CUMS treated with FLX (10 mg/kg). During the CUMS protocol the coat state and the sucrose preference test were evaluated weekly. Once the treatments were finalized, mice were exposed to forced swimming tests. (**B**) rTMS Mag Pro Cool-40 coil for use in rodents. The red arrow points to the induction point (the point of the coil with the highest magnetic field intensity). (**C**) Front and side diagram of the Cool-40 coil showing the maximum rate of change in the magnetic field strength in kT/s, according to the distance (millimeters, mm) from the induction point. (**D**) Schematic drawing showing the coil placement on the mouse head. Also, it shows that the induction point is located just above the FC. (**E**) Representative photographs in the frontal and lateral view of the application of rTMS in mice.

**Figure 2 cells-12-02062-f002:**
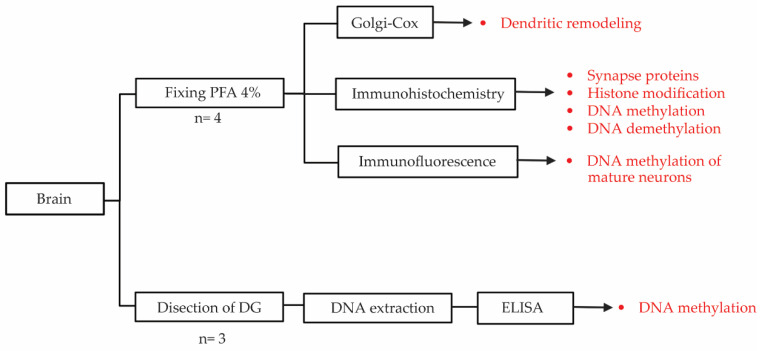
Flow chart of sample processing, showing the distribution of the brain hemispheres used for each technique. The arrows on the right indicate the parameters analyzed (shown in red).

**Figure 3 cells-12-02062-f003:**
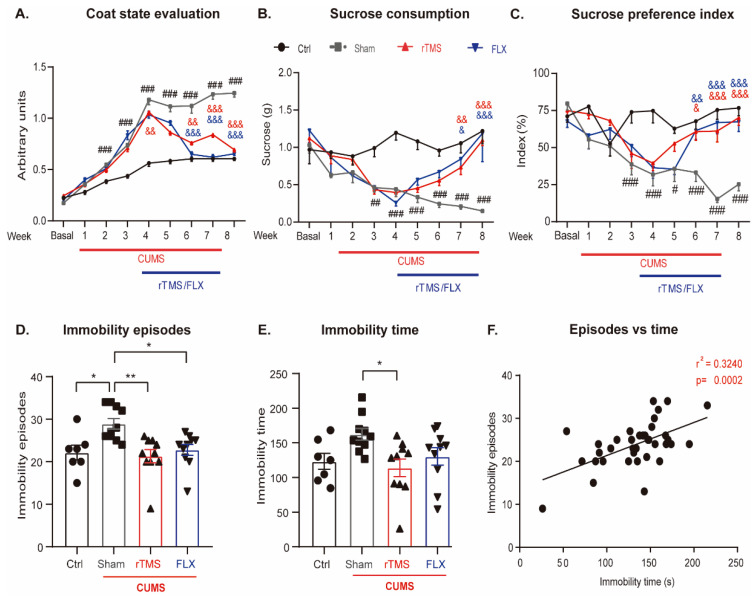
Repetitive transcranial magnetic stimulation reverses depressive-like behaviors generated in mice exposed to chronic unpredictable mild stress. Experimental groups consisted of the control without stress (Ctrl), sham exposed to chronic stress (CUMS), and mice exposed to CUMS treated with rTMS or FLX (10 mg/Kg). (**A**) The coat state was evaluated weekly. The coat deterioration was evident after the third week following CUMS initiation, but rTMS or FLX (10 mg/kg) reversed it (which was determined using the two-way repeated measures ANOVA test followed by the Bonferroni post hoc test). (**B**) Sucrose consumed in the sucrose preference test reveals similar effects to those observed in the coat state of mice exposed to CUMS. Also, rTMS and FLX improved sucrose consumption (which was determined using the two-way repeated measures ANOVA test followed by the Bonferroni post hoc test). (**C**) In the sucrose preference index, a behavior like sucrose consumption is observed since stress decreased this parameter from week 3 until the end of the CUMS protocol. Meanwhile, in mice treated with rTMS and FLX, an increase is observed, from week 6 to week 8 (which was determined using the two-way repeated measures ANOVA test followed by the Bonferroni post hoc test). (**D**) Number of immobility episodes observed in the forced swim test. Stress increased the number of immobility episodes, while rTMS and FLX reversed this effect (One-way ANOVA. Bonferroni post hoc). (**E**) Immobility time in the forced swim test. Stress tends to increase the immobility time, while rTMS reverses this effect, and Flx tends to do so (which was determined using the one-way ANOVA followed by the Bonferroni post hoc test). (**F**) Linear regression between the immobility episodes and the immobility time in the forced swim test (which was assessed using the Pearson’s correlation coefficient. N = 37). The panels (**A**–**C**) (#) symbols represent significant differences between the CUMS-treated and Ctrl groups (#: *p* ≤ 0.05, ##: *p* ≤ 0.005, ###: *p* ≤ 0.001). Also, (&) symbols represent the significant differences found among the rTMS (red symbol) or FLX (blue symbol) groups versus the sham groups after the evaluations were performed every week. (&: *p* ≤ 0.05, &&: *p* ≤ 0.005, &&&: *p* ≤ 0.001). The red bar below indicates the time exposed to the CUMS protocol (except for the Ctrl group), and the blue bar indicates the weeks where rTMS or FLX were applied in their respective groups. In panels (**D**,**E**), symbols represent a significant difference between groups (* *p* ≤ 0.05, ** *p* ≤ 0.005). The red bars indicate the groups that were exposed to the CUMS protocol. Error bars represent the standard error of the mean. n = 7–10.

**Figure 4 cells-12-02062-f004:**
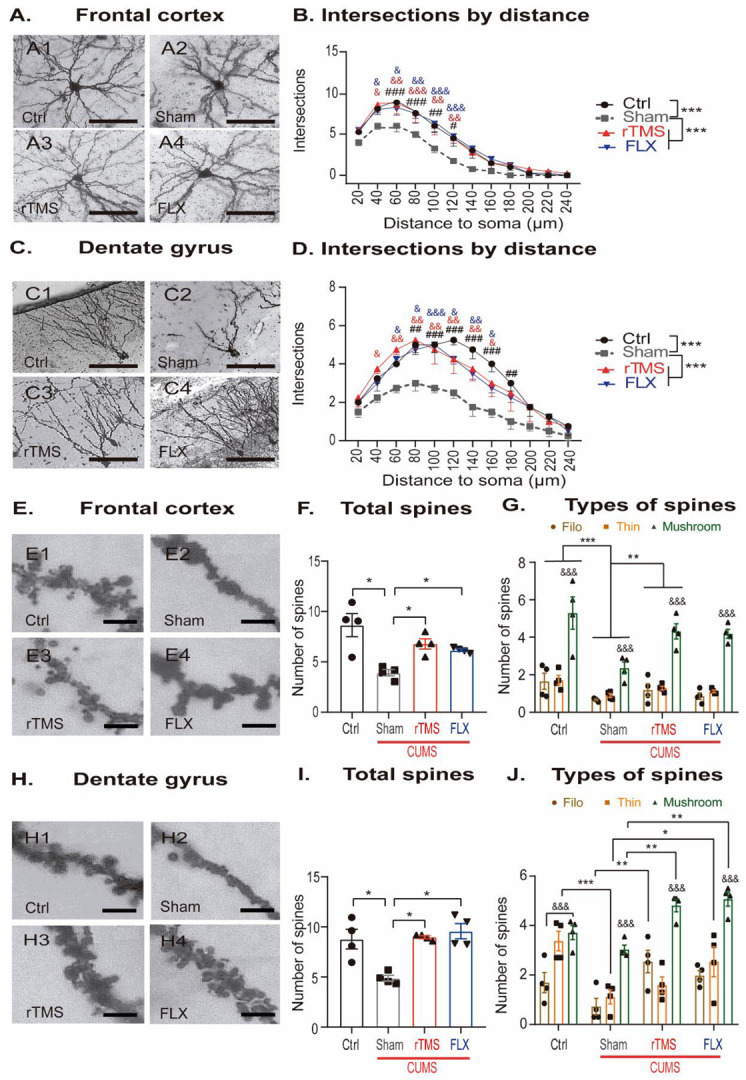
Repetitive transcranial magnetic stimulation reverses the alterations of dendritic complexity and the morphology of dendritic spines in chronically stressed mice. (**A**) Representative micrographs of the pyramidal neurons of layer II/III of the frontal cortex (FC) (A1–4) of the experimental groups Ctrl (A1), sham (A2), rTMS (A3), and FLX (10 mg/Kg) (A4). Scale bar = 100 µm. (**B**) Dendritic complexity was measured as the number of intersections observed at particular distances (or up to 20 μm less) from the soma. Stress reduces the number of intersections in the FC. The application of rTMS or FLX reverses this effect (which was determined using the two-way repeated measures ANOVA followed by the Bonferroni post hoc test). (**C**) Representative micrographs of the granular neurons of the dentate gyrus (DG) (C1–4) of the experimental groups Ctrl (A1, C1), sham (A2, C2), rTMS, (A3, C3) and FLX (10 mg/Kg) (A4, C4). Scale bar = 100 µm. (**D**) Dendritic complexity was measured as the number of intersections observed at particular distances (up to 20 μm less) from the soma. Stress reduces the number of intersections in the FC and the DG, respectively. Applying rTMS or FLX reverses this effect in both regions (which was determined with the two-way repeated measures ANOVA followed by the Bonferroni post hoc test). (**E**) Representative micrographs of the dendritic spines of the pyramidal neurons of layer II/III of the FC (E1–4) of the experimental groups Ctrl (E1), sham (E2), rTMS (E3), and FLX (10 mg/Kg) (E4). Scale bar = 2.5 µm. (**F**) Stress decreases the number of dendritic spines in the FC and rTMS, and FLX reverses this effect (which were determined with the one-way on ranks ANOVA followed by the Student–Newman–Keuls post hoc method. (**G**) Quantification of the DSs according to their morphology in the FC. Stress reduces the number of DS, independent of their morphology; rTMS and FLX reversed this effect (which was determined using the two-way ANOVA followed by the Bonferroni post hoc test). (**H**) Representative micrographs of the granular neurons in the DG (H1–4) of the experimental groups Ctrl (H1), sham (H2), rTMS (H3), and FLX (10 mg/Kg) (H4). (**I**) Stress decreases the number of dendritic spines in the DG and rTMS, and FLX reverses this effect (which was determined using the one-way ANOVA followed by the Bonferroni post hoc test). (**J**) Quantification of the DSs according to their morphology in the DG. Stress significantly decreases the density of the thin DS type, while rTMS significantly increases the density of the filo and mushroom DSs, and FLX significantly increases the thin and mushroom DSs (which were determined using the two-way ANOVA followed by the Bonferroni post hoc test). In panels (**B**,**D**) the (#) symbol represents significant differences between the CUMS and Ctrl groups along the dendritic trees (#: *p* ≤ 0.05, ##: *p* ≤ 0.005, ###: *p* ≤ 0.001), while the (&) symbol represents significant differences found among the rTMS (red symbol) or FLX (blue symbol) groups versus the sham groups along the dendritic trees (&: *p* ≤ 0.05, &&: *p* ≤ 0.005, &&&: *p* ≤ 0.001). Also, the (*) symbol represents the significant difference between groups without considering the sham or other groups at particular distances from the soma (*** *p* ≤ 0.001). In panels (**F**,**I**), (*) symbols represent a significant difference between groups (* *p* ≤ 0.05). In panels (**G**,**J**), the form ● indicate the number of DS type filo, ■ indicate of DS type thin and ▲ indicate of DS type mush, (&) symbols represents a significant difference between the DS types in the same group (&&&: *p* ≤ 0.001). In panel (**G**), (*) symbols represent significant differences between the groups independent of the DS type (**: *p* ≤ 0.05, ***: *p* ≤ 0.001), and in panel (**J**), (*) symbols represent significant differences between the groups considering the DS-type of the DS (*: *p* ≤ 0.05, **: *p* ≤ 0.001, *** *p* ≤ 0.001). Error bars represent the standard error of the mean *n* = 4.

**Figure 5 cells-12-02062-f005:**
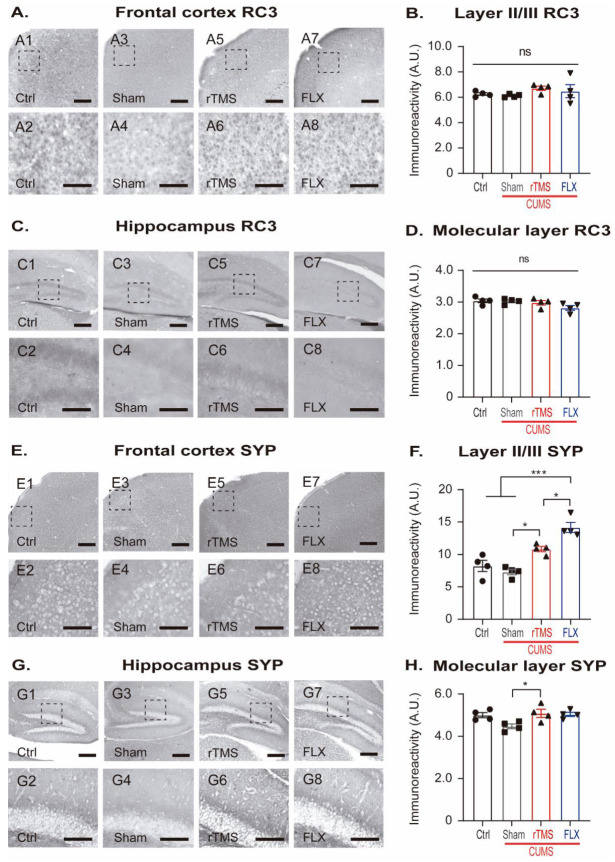
Repetitive transcranial magnetic stimulation and fluoxetine modify the immunoreactivity of synaptophysin without affecting the immunoreactivity of neurogranin. (**A**) Representative micrographs at 10× (scale bar = 400 µm) and 40× (scale bar = 100 µm) of neurogranin (RC3) immunoreactivity in layer II/III of the frontal cortex (FC) (A1–8) of the experimental groups Ctrl (A1, 2), sham (A3, 4), rTMS (A5, 6), and FLX (10 mg/Kg) (A7, 8). (**B**) The CUMS, rTMS, and FLX applications did not modify the RC3 immunoreactivity in the FC (one-way on ranks ANOVA). (**C**) Representative micrographs at 10× (scale bar = 400 µm) and 40× (scale bar = 100 µm) of neurogranin (RC3) immunoreactivity in the molecular layer (ML) of the hippocampus of the experimental groups Ctrl (C1, 2), sham (C3, 4), rTMS (C5, 6), and FLX (10 mg/Kg) (C7, 8). (**D**) The CUMS, rTMS, and FLX applications did not modify the RC3 immunoreactivity in the ML (one-way ANOVA). (**E**) Representative micrographs at 10× (scale bar = 400 µm) and 40× (scale bar = 100 µm) of synaptophysin (SYP) immunolabeling in layer II/III of the FC (E1–8) of the experimental groups Ctrl (E1, 2), sham (E3, 4), rTMS (E5, 6), and FLX (10 mg/Kg) (E7, 8). (**F**) Chronic stress did not modify the SYP immunoreactivity in the FC, but rTMS and FLX increased the SYP immunoreactivity in this region (which were determined using the one-way ANOVA followed by the Bonferroni post hoc test). (**G**) Representative micrographs at 10× (scale bar = 400 µm) and 40× (scale bar = 100 µm) of synaptophysin (SYP) immunolabeling in the ML of the hippocampus (G1–8) of the experimental groups Ctrl (G1, 2), sham (G3, 4), rTMS (G5, 6), and FLX (10 mg/Kg) (G7, 8). (**H**) In the ML, rTMS increased the immunoreactivity of SYP compared to the sham group (which was determined using the one-way ANOVA followed by the Bonferroni post hoc test). In all panels (*) represent a statistically significant difference (* *p* ≤ 0.05, *** *p* ≤ 0.001). Error bars represent the standard error of the mean. *n* = 4. A.U—arbitrary units.

**Figure 6 cells-12-02062-f006:**
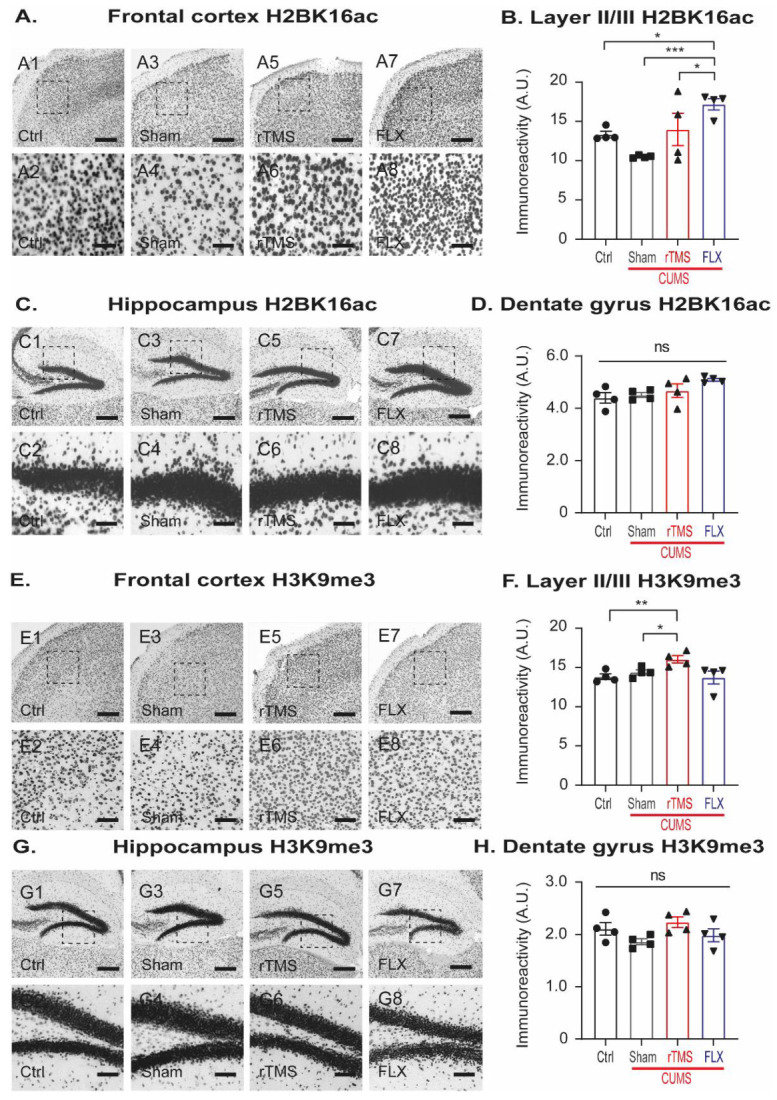
Differential effects of repetitive transcranial magnetic stimulation and fluoxetine on the immunoreactivity of the histone H2B (K16) acetylation and histone H3 trimethylation. (**A**) Representative micrographs at 10× (scale bar = 400 µm) and 40× (scale bar = 100 µm) of the histone H2B acetylation in lysine 16 (H2BK16ac) immunolabeling in layer II/III of the frontal cortex (FC) (A1–8) of the experimental groups Ctrl (A1, 2), sham (A3, 4), rTMS (A5, 6), and FLX (10 mg/Kg) (A7, 8). (**B**) FLX increased the H2BK16ac immunoreactivity compared with the Ctrl, sham, and rTMS groups in the FC (which was determined using the one-way ANOVA followed by the Bonferroni post hoc test). (**C**) Representative micrographs at 10× (scale bar = 400 µm) and 40× (scale bar = 100 µm) of the histone H2B acetylation in lysine 16 (H2BK16ac) immunolabeling in the dentate gyrus (DG) of the hippocampus (C1–8) of the experimental groups Ctrl (C1, 2), sham (C3, 4), rTMS (C5, 6), and FLX (10 mg/Kg) (C7, 8). (**D**) Stress and the therapeutic interventions did not modify the H2BK16ac immunoreactivity in the DG (one-way ANOVA). (**E**) Representative micrographs at 10× (scale bar = 400 µm) and 40× (scale bar = 100 µm) of the histone H3 trimethylation in lysine 9 (H3K9me3) immunolabeling in the layer II/III of the FC (E1–8) of the experimental groups Ctrl (E1, 2), sham (E3, 4), rTMS (E5, 6), and FLX (10 mg/Kg) (E7, 8). (**F**) Stress does not modify the H3K9me3 immunoreactivity but the rTMS increased it in the FC (which was determined using the one-way ANOVA followed by the Bonferroni post hoc test). (**G**) Representative micrographs at 10× (scale bar = 400 µm) and 40× (scale bar = 100 µm) of the histone H3 trimethylation in lysine 9 (H3K9me3) immunolabeling in the ML of the hippocampus (G1–8) of the experimental groups Ctrl (G1, 2), sham (G3, 4), rTMS (G5, 6), and FLX (10 mg/Kg) (G7, 8). (**H**) Stress and the therapeutic interventions do not modify the H3K9me3 in the DG (one-way ANOVA). In all panels * represents a significant difference between groups (* *p* ≤ 0.05, ** *p* ≤ 0.005, *** *p* ≤ 0.001). Error bars represent the standard error of the mean. *n* = 4. A.U—arbitrary units.

**Figure 7 cells-12-02062-f007:**
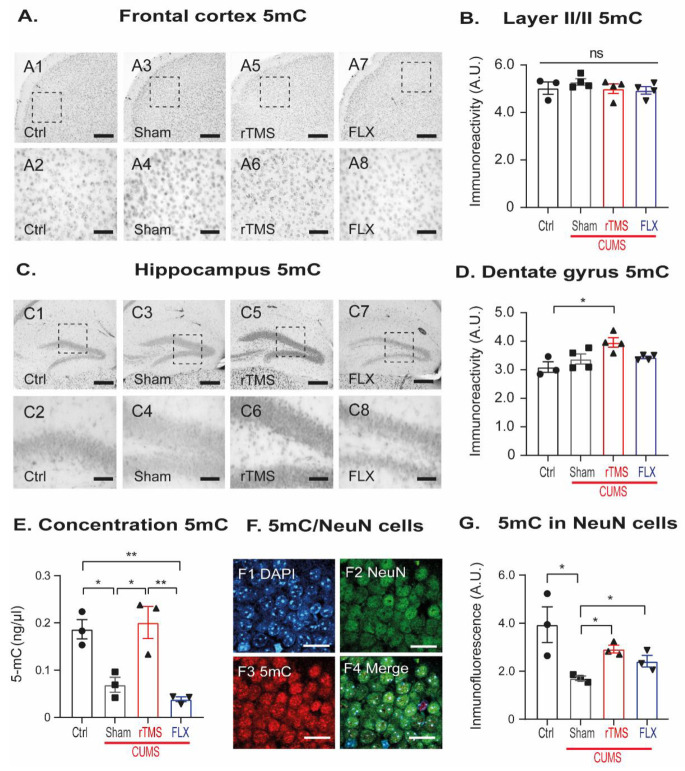
Repetitive transcranial magnetic stimulation and fluoxetine differentially modify the expression of 5-methyl-cytosine in the hippocampus, but not in the frontal cortex, in chronically stressed mice. (**A**) Representative micrographs at 10× (scale bar = 400 µm) and 40× (scale bar = 100 µm) of 5mC immunolabeling in layer II/III of the frontal cortex (FC) (A1–8). Experimental groups were control without stress (Ctrl) (A1, C1), sham exposed to chronic stress (CUMS) (A2, C2), mice exposed to CUMS treated with 5 Hz rTMS (A3, C3), and FLX (10 mg/Kg). (**B**) Stress and the therapeutic interventions with rTMS or FLX did not modify the immunoreactivity of 5mC in the FC (One-way ANOVA). (**C**) Representative micrographs at 10× (scale bar = 400 µm) and 40× (scale bar = 100 µm) of 5mC immunolabeling in the dentate gyrus (DG) of the hippocampus (C1–8). Experimental groups were control without stress (Ctrl) (A1, C1), sham exposed to chronic stress (CUMS) (A2, C2), mice exposed to CUMS treated with 5 Hz rTMS (A3, C3), and FLX (10 mg/Kg). (**D**) rTMS, but not FLX, increased 5mC immunoreactivity in DG (determined using one-way ANOVA followed by the Bonferroni post hoc test). (**E**) Stress decreased the concentrations of 5mC and rTMS, but not FLX, which reversed this effect in the DG (determined using the one-way ANOVA followed by the Bonferroni post hoc test). (**F**) Representative micrographs of 5mC/NeuN/DAPI-positive cells in the DG. Blue: DAPI (F1), green: NeuN (F2), red: 5mC (F3), and merge: 5mC/NeuN/DAPI (F4) (scale bar = 20 µm). (**G**) Stress reduced the immunofluorescence of 5mC in the mature neurons of the DG, but rTMS reversed this effect (which was determined using the one-way ANOVA on ranks followed by the Student–Newman–Keuls test post hoc test). In all panels * represents a significant difference between groups (* *p* ≤ 0.05, ** *p* ≤ 0.005). Error bars represent the standard error of the mean. *n* = 3–4. A.U—arbitrary units.

**Figure 8 cells-12-02062-f008:**
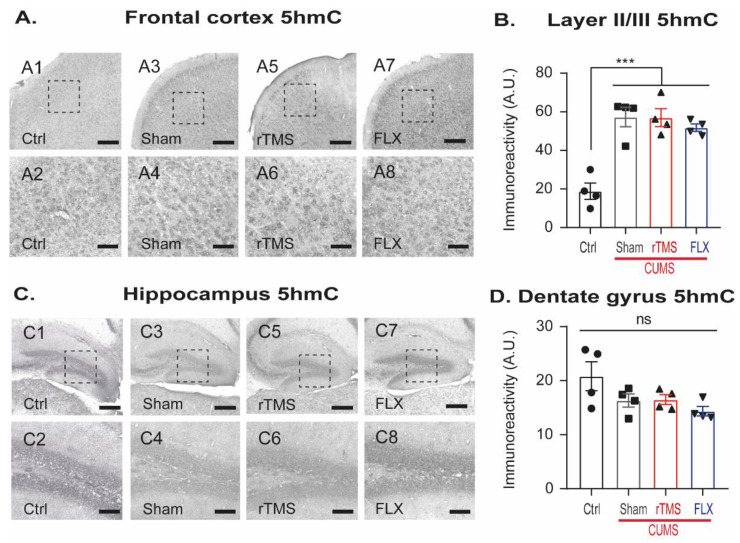
Repetitive transcranial magnetic stimulation and fluoxetine did not increase the levels of 5-hydroxy-methyl-cytosine generated by chronic stress. (**A**) Representative micrographs at 10× (scale bar = 400 µm) and 40× (scale bar = 100 µm) of 5-hydroxy-methyl-cytosine (5hmC) immunolabeling in layer II/III of the frontal cortex (FC) (A1–8). Experimental groups were Ctrl (A1, 2), sham (A3, 4), rTMS (A5, 6), and FLX (A7, 8). (**B**) CUMS increased 5hmC immunoreactivity in the FC, while rTMS or FLX did not reverse this effect (which was determined using the one-way ANOVA followed by the Bonferroni *post hoc* test). (**C**) Representative micrographs at 10× (scale bar = 400 µm) and 40× (scale bar = 100 µm) of 5-hydroxy-methyl-cytosine immunolabeling in the dentate gyrus (DG) of the hippocampus (C1–8). Experimental groups were Ctrl (C1, 2), sham (C3, 4), rTMS (C5, 6), and FLX (C7, 8). (**D**) Neither CUMS, rTMS, or FLX were able to modify 5hmC immunoreactivity in the DG of the hippocampus (one-way ANOVA on ranks). In all panels * represents a significant difference between groups (*** *p* ≤ 0.001). *n* = 4. A.U—arbitrary units.

**Figure 9 cells-12-02062-f009:**
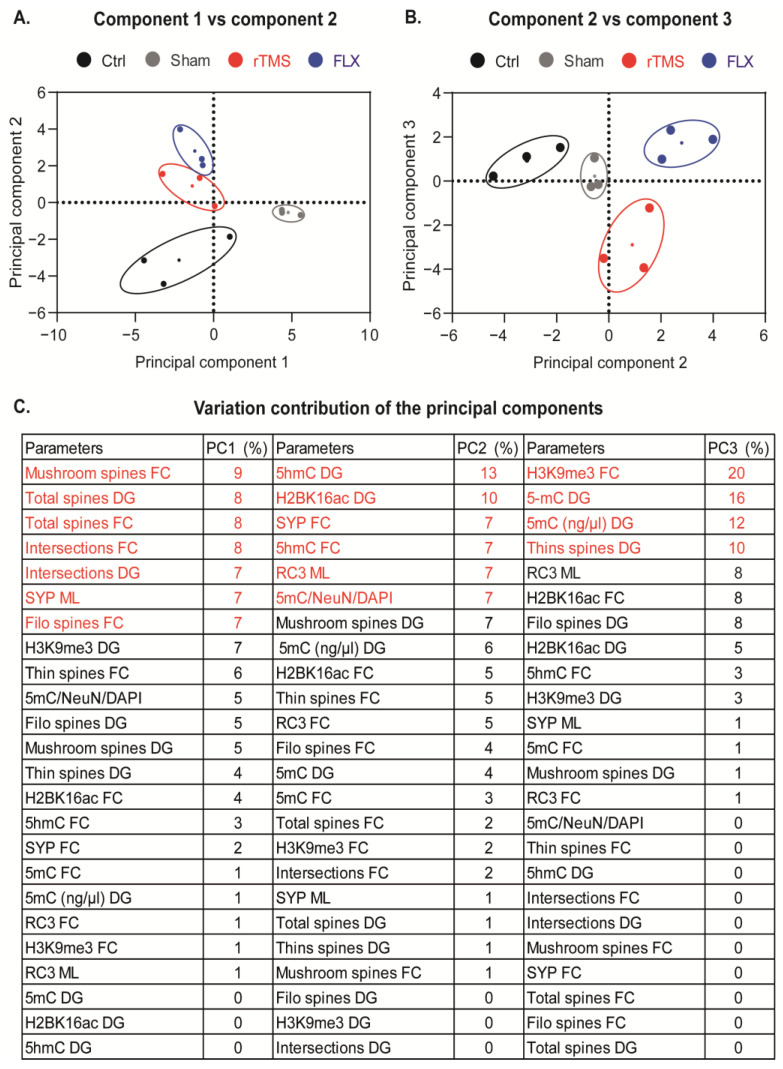
Principal component analysis of the continuous variables of dendritic remodeling, synapse proteins, and epigenetic markers. (**A**) PC score plot of component 1 vs. component 2. Component 1 explains the effects of stress and shows the similarities between the Ctrl group and the antidepressant treatment groups. (**B**) PC score plot of component 2 vs. component 3. Component 3 shows the principal variables that explain the effects of rTMS and FLX. (**C**) Table of contribution of the variance values of the three principal components. Red indicates the main parameters with a higher percentage that explain the variance by component. The cut-off points for choosing these parameters were that the sum of all of them should explain 50% of the variance. The Kaiser criterion was used to determine the number of principal components to be retained, selecting those with eigenvalues greater than 1. *n* = 3.

**Table 1 cells-12-02062-t001:** Stressors used in the CUMS protocol.

Stressor	Time
Empty box	4 h
Movement restriction	1 h
Rotating box	1 h
Continuous light	24 h
Cold (5 °C)	20 min
White noise	12 h
Strobe light	4 h
Tilted box (45°)	3 h
Wet bed	12 h
Overcrowding	4 h
Predator odor	3 h
Food deprivation	18 h
Wet box	1 h 30 min

## Data Availability

Data will be made available on request.

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
