# Peer review of "Repetitive Transcranial Magnetic Stimulation Reduces Depressive-like Behaviors, Modifies Dendritic Plasticity, and Generates Global Epigenetic Changes in the Frontal Cortex and Hippocampus in a Rodent Model of Chronic Stress"

_cells, 2023, doi:10.3390/cells12162062_

Round 1
Reviewer 1 Report
This research shows that repetitive transcranial magnetic stimulation (rTMS, 5 Hz), similarly to chronic fluoxetine administration, reverses behavioural alterations generated by chronic stress in female mice. Of particular interest is a first approximation of the epigenetic effects of rTMS in an animal model of depressive-like behaviors. Overall this is a well-designed study and provides very interesting results. With some minor editing to the manuscript this paper is deserving of publication.
My suggestions/some weaknesses:
- - Fig. 2D: Looking at the fig, 2 D, it seems that stress does not increase the immobility time. We can only observed some tendency. Similarly, when we compare FLX-group and Sham-group.
- - Please provide an explanation for the choice of animal sex – female.
- - why did you measure two parameters in FST? Is there a difference between these parameters with regard to depressive behavior?
Author Response
REVIEWER 1: This research shows that repetitive transcranial magnetic stimulation (rTMS, 5 Hz), similarly to chronic fluoxetine administration, reverses behavioral alterations generated by chronic stress in female mice. Of particular interest is a first approximation of the epigenetic effects of rTMS in an animal model of depressive-like behaviors. Overall, this is a well-designed study and provides very interesting results. With some minor editing to the manuscript this paper is deserving of publication.
My suggestions/some weaknesses:
- - Fig. 2D: Looking at the fig, 2 D, it seems that stress does not increase the immobility time. We can only observe some tendency. Similarly, when we compare FLX-group and Sham-group.
A: In our work, we observed a tendency to increase the immobility time in the Sham group compared to the Ctrl group and a tendency to decrease this effect in the Flx group compared to the Sham group (Figure 2D). However, these changes are not significant (p=0.185 and p=0.299, respectively). We believe this is due to the dispersion of the data of these groups, which prevents us from detecting significant changes with the one-way ANOVA and Bonferroni post hoc test. However, when we applied the t-test to compare the Ctrl vs Sham group and the Sham vs Flx group, we found significant changes (p=0.013 and p=0.486, respectively). In addition, the data on the number of immobility episodes (Figure 2C) suggest that the stress protocol induces despair and that Flx and rTMS reverse this effect.
- - Please provide an explanation for the choice of animal sex – female.
We chose to work with female Balb/C mice instead of males for the following reasons:
- The incidence of major depressive disorder is more frequent in females than males, with a ratio of 2:1[1].
- Limited research uses chronic stress models in females, as this research is predominantly conducted in males [2].
- In females, the effects of stress in animal models are more readily observable. It has been reported that females have greater activation of the hypothalamic-pituitary-adrenal (HPA) axis [3] and an increase in the methylation of the corticotropin-releasing factor (CRF) gene compared to males exposed to stress [4]
For these reasons, we chose to work with females in our study. However, future studies need to include both females and males to evaluate the influence of sex on the neuroplastic and epigenetic changes associated with the antidepressant effect of rTMS.
- - why did you measure two parameters in FST? Is there a difference between these parameters with regard to depressive behavior?
A: In our study, we measured the total immobility time and the number of immobility episodes. This parameter provides additional information on whether the immobility was continuous or intermittent. The number of immobility episodes can indicate the transition from a passive state (immobility) to an active state (active swimming) in mice during the forced swimming test. Both the time and number of immobility episodes are used to evaluate despair behavior [5][6][7], although the total immobility time is the most commonly reported parameter for this behavior [8]. In our study, these two parameters were positively correlated (Figure 1) and complemented each other in confirming the effect of treatments on despair behavior.
Figure 3. Repetitive transcranial magnetic stimulation reverses depressive-like behaviors generated in mice exposed to chronic unpredictable mild stress. Experimental groups were control without stress (Ctrl), Sham exposed to chronic stress (CUMS), mice exposed to CUMS treated with rTMS or FLX (10 mg/Kg). A. The coat state was evaluated weekly. The coat deterioration was evident after the third week of the CUMS, but rTMS or FLX (10 mg/kg) reversed it (Two-way repeated measures ANOVA. Bonferroni test post hoc). B. Sucrose consumed in the sucrose preference test reveals similar effects to those observed in the coat state of mice exposed to the CUMS. Also, rTMS and FLX improved sucrose consumption (Two-way repeated measures ANOVA. Bonferroni test post hoc). C. In the sucrose preference index, a behavior like sucrose consumption is observed since stress decreased this parameter from week 3 until the end of the CUMS protocol. While in mice treated with rTMS and FLX, an increase is observed, from week 6 to week 8, (Two-way repeated measures ANOVA. Bonferroni test post hoc). D. Number of immobility episodes in the forced swim test. Stress increased immobility episodes, and rTMS and FLX reversed this effect (One-way ANOVA. Bonferroni post hoc). E. Immobility time in the forced swim test. Stress tends to increase the immobility time, rTMS reverses this effect, and Flx tends to do so (One-way ANOVA. Bonferroni post hoc). F. Linear regression between the immobility episodes and immobility time in the forced swim test. (Correlation Pearson). The Panel A, B, and C (#) symbols represent significant differences between the CUMS and the Ctrl groups (#: p ≤ 0.05, ##: p ≤ 0.005, ###: p ≤ 0.001). Also, (&) symbols represent significant differences found among the rTMS (red symbol) or FLX (blue symbol) groups versus Sham groups after the evaluations were done every week. (&&: p ≤ 0.005, &&&: p ≤ 0.001). The red bar below indicates the time exposed to the CUMS (except for the Ctrl group), and the blue bar indicates the weeks where rTMS or FLX were applied in their respective groups. In panels D and E, symbols represent a significant difference between groups (p ≤ .05, * p ≤ 0.005). In panel F (**), symbols represent a significative correlation (* p ≤ 0.005). N= 37. The red bars indicate the groups that were exposed to the CUMS. Error bars represent the standard error of the mean. n= 7-10.
We want to thank this reviewer's observations and hope that our answers satisfactorily respond to the questions of this reviewer.

Reviewer 2 Report
In this MS authors showed that repeated transcranial magnetic stimulation and fluoxetine (used as positive control) induced similar behavioral and neuroplastic effects, although distinct epigenetic mechanisms. This is a wonderful work, very well planned, well designed, and well executed. The manuscript is quite long, but it is full of information, scientifically based, which certainly will be of interest to the scientific literature.
Reading the manuscript I listed some minor details that are described below:
1- Authors used a lot of abbreviations in the entire MS. I suggest reducing it, for example: avoid abbreviating behavioral tests or other words that are available in limited sessions of the MS. In the abstract, DG is not written in full.
2- Sucrose preference test is a widely used animal model to measure anedonia in rodents. However, in the methods, authors do not describe how they calculate sucrose consumption. Additionally, in the description of this test, page 6, line 5, please check if “prior to” is correctly used.
3- Additionally, when they display the amount of sucrose consumption without showing the amount of water consumption, we cannot be sure that really animals prefer sucrose instead of water (Fig 2B). Please check the error lines in sham group. Error lines are available in all groups except for sham.
4- Please mention the water temperature used in the Forced Swimming Test. It is relevant information for data replicability.
Author Response
REVIEWER 2: In this MS authors showed that repeated transcranial magnetic stimulation and fluoxetine (used as positive control) induced similar behavioral and neuroplastic effects, although distinct epigenetic mechanisms. This is a wonderful work, very well planned, well designed, and well executed. The manuscript is quite long, but it is full of information, scientifically based, which certainly will be of interest to the scientific literature. Reading the manuscript, I listed some minor details that are described below:
R2Q1. Authors used a lot of abbreviations in the entire MS. I suggest reducing it, for example: avoid abbreviating behavioral tests or other words that are available in limited sessions of the MS. In the abstract, DG is not written in full.
A: We appreciate this observation. In the new version of our manuscript, we decrease the number of abbreviations.
R2Q2. Sucrose preference test is a widely used animal model to measure anhedonia in rodents. However, in the methods, authors do not describe how they calculate sucrose consumption. Additionally, in the description of this test, page 6, line 5, please check if “prior to” is correctly used.
A: We apologize for not including the graph for the sucrose preference index. To evaluate this parameter, we calculated the sucrose preference index as follows: (Sucrose consumed x (sucrose consumed + natural water consumed)/100)). In the revised version of our manuscript, we have added a histogram to Figure 3, panel C. Information about this panel can be found in the figure legend.
R2Q3. Additionally, when they display the amount of sucrose consumption without showing the amount of water consumption, we cannot be sure that really animals prefer sucrose instead of water (Fig 2B). Please check the error lines in the sham group. Error lines are available in all groups except for sham.
We agree with this observation. Thus, we included the sucrose preference index in which water consumption is considered. Please see the answer to the previous observation (R2Q2).
Regarding the error lines in the sucrose consumption, we found a little variation among the data in weeks 7 and 8, which is why we cannot see the error bars.
R2Q4. Please mention the water temperature used in the Forced Swimming Test. It is relevant information for data replicability.
A: The information required was included in the materials and methods section. Please see the fourth line of subsection 2.6.3. in which is written: “25 ± 1 ÌŠ C.”.
We want to thank this reviewer's observations and hope that our answers satisfactorily respond to the questions of this reviewer.

Reviewer 3 Report
Repetitive transcranial magnetic stimulation (rTMS) is a valuable emerging option for treating depression. It is important to explore mechanisms by which rTMS works. This paper is a worthwhile examination of the effects of rTMS on neuronal structures and epigenetics in mice.
The writing starts off clear. However, it later displays many errors. There are a huge number of careless mistakes throughout most of the paper, and I tried to list them below.
Line 59: Change "intermediated" to "intermediate".
Line 62: Change "desoxyribonucleic" to "deoxyribonucleic".
Line 74: Change "DNA." to "DNA".
Line 75: Change "," to ".".
Line 84: Change "generate different alterations in the brain ranging from the" to "changes brain", or something like that.
Line 86: Delete "the" before BDNF, unless you're discussing a particular subset of BDNF expression (in which case you should specify which BDNF expression is increased).
Line 88: Move the comma to before "and".
Line 89: Change "the long-term potentiation" to "long-term potentiation".
Line 105: Change "Balb/C" to "BALB/c" here and throughout the paper.
Line 109: Change "along" to "during the", or something like that.
Lines 123-125: In this section, explain more clearly how you applied CUMS. You gave 1-3 stressors per day, but how did you choose the number of stressors to apply each day? Was that random? Once the number was chosen, how did you choose a stressor from Table 1? Was that also random? Some stressors last 12-24 hours, thus they can't be applied 2 or 3 times a day. How did you deal with that? Or did you choose one stressor each day and then apply that same stressor all day?
Line 129: Change "USA)" to "USA". Change "wide" to "width".
Line 130: Change "amplitude/magnetic field strength" to "change in magnetic field strength per time".
Line 133-4: Change "10%, increasing every 2%" to "10% of maximum, increasing by 2% of maximum".
Line 140: Instead of "0" to indicate degrees, use the actual degree symbol.
Lines 155-158: You listed the groups twice.
Line 156: Change "Kg" to "kg" throughout the paper.
Line 162: Change "magnetic field strength" to "maximum rate of change in magnetic field strength".
Lines 186-192: Why did you restrict food and water after the SPT? Why did you change the protocol from that in reference 42?
Line 189: Change "Week basal" to "basal week".
Line 199: Is the phrase in parentheses an example? If so, indicate it.
Line 203: What do you mean by "half": every other brain, or one hemisphere of each brain? I assume you mean the former. If so, one way to clarify this would be to say "about half". Anyway, "about half" is more correct than just "half" because some groups had odd numbers of mice (for example, n=7).
Line 209: Spell out what 5mC is the first time you mention it.
Line 230: What do you mean by "rate"?
Line 233 and throughout the paper: Delete "the" before "Sholl".
Line 235-238: It's confusing that you describe killing both here and section 2.7. Consolidate these and only describe killing once. You might add a small flowchart indicating the steps and numbers of mice: 7-10 mice per experimental group, which were all killed and brains removed, of which about half were used to extract DNA, one was Golgi Cox stained, and the remainder were used for immunohistochemistry. It took me quite a while to piece together the above flowchart from the text.
Line 258: Change "Merk" to "Merck". I couldn't find "Newmont medium" anywhere on the Merck SigmaAldrich website or even anywhere on the internet. You may list the catalog number in the paper.
Line 260: Delete "Images of " because you say "were imaged" later.
Line 262: Add a period at the end of the sentence (after "v. 3.4.0").
Lines 266-271: What do you mean by "quadrants"? Is the microscope field divided into 4 parts? Why quadrants instead of whole fields?
Line 273: You skipped section 2.10.
Line 273: Add a period at the end, to be consistent with your other subheadings.
Sections 2.9-2.11: The section headings are confusing. You may rearrange and retitle these sections, perhaps so that there is one heading on immunohistochemistry, with subheadings for tissue processing for immunohistochemistry, synapse proteins, and 5mC.
Line 293: Change the period to a comma.
Line 297: Change "Glomax discover, Promega." to "GloMax Discover, Promega,".
Line 302: Referring to data as ordered in distributions doesn't seem quite correct. Maybe you're referring to the ordering of data in nonparametric tests. But I think it makes more sense to replace "in which our data were ordered" with "of our data".
Lines 311-6: I have no experience with PCA, so I can't evaluate this.
Line 318: Change "and" to "or".
Lines 320-2: This sentence has some redundancies. You may change it to something like: "The coat deteriorated in chronically stressed mice compared to control mice (p<0.001)."
Line 323: The 4th week doesn't show the worst CS. It looks like the 8th week is worse. You may change "the highest" to "a plateau in the" and delete the subsequent sentence.
Line 329 and later: You don't need 5 significant digits in the F values. 3 seems plenty. By analogy, if you asked me my age, you'd think I was weird if I said I'm 58.284 years old. You probably only want to hear 2 significant digits for my age. Throughout the paper, reduce the number of significant digits to something reasonable.
Line 331 and later: Change "X" to "×".
Line 331: Add a close parenthesis.
Line 335 and later: "Revert" is an intransitive verb (https://www.merriam-webster.com/dictionary/revert). Change "reverted" to "reversed".
Line 335: It's surprising that rTMS and FLX are not significantly different than controls in week 6. Looking at the error bars in the graph, they look different. But maybe looks are deceiving.
Line 336: Move "mice" before "without".
Line 339: Add a close parenthesis.
Line 345: Change "number" to "increase in number".
Lines 345-6: Why do you show 2 p-values?
Line 347: Why do you show 2 p-values?
Figure 2D: In the vertical axis label, delete "(s)".
Line 352: Change " Control" to "control".
Line 355: Change "and" to "or".
Line 355 and throughout the paper: Put the adjective before the noun. Change “ANOVA two-way repeated measurement” to “Two-way repeated measures ANOVA”. From this point onward in the paper, you often reversed the order of the name of ANOVA tests. For example: “ANOVA one-way”. It should be “one-way ANOVA”. Find these throughout the paper and correct them.
Figure 2: If you want to show significance with different symbols in each part of the figure, you should state the explanation for each symbol you use in the legend for each part of the figure (A, B, C, and D). Alternatively, you can devise symbols that work for all parts of the figure; then you need to explain them only once in the legend: at the end of the first part of the legend, before the specific legend for part A.
Lines 355-6: Why do you say "ANOVA two-way repeated measurement. Bonferroni test post hoc." without making any quantitative statement about statistical results in Figure 2A?
Line 356: Delete ",".
Line 359: Change "along the time exposure to the CUMS versus the Ctrl group" to "between the CUMS and the Ctrl groups".
Lines 360-1: Explain what levels of significance && and &&& indicate.
Line 366: Stress does not significantly increase the immobility time. It looks like a trend, though.
Lines 366-7: FLX does not significantly reverse the effect. It may be a trend.
Line 370: Change "Line" to "Error".
Line 376: Change "dendrite trees in" to "dendritic trees of".
Line 380: Add "the" before "control".
Line 381: Change "from 20 to 140 micrometers related to the soma" to "at distances of 20 to 140 micrometers from the soma".
Lines 381-2: Are these p values correct? The figure caption says p<0.001.
Line 385: Add a close parenthesis.
Line 386: It's not trees that are impregnated with dendrites, so change "dendrite-impregnated trees in" to "dendritic trees of". I think it's not necessary to state again here that you're looking only at impregnated cells; you've already described this in the Methods.
Line 392: Change "from 60 to 180 micrometers related to the soma" to "at distances of 60 to 180 micrometers from the soma".
Line 395: Add a close parenthesis.
Line 396: Change "concerning the distance" to "at different distances".
Lines 400-3: Define the q values you use. Usually q values are between 0 and 1. Explain the statistical analysis you used, and how you examined significance.
Line 407: Change "The FC confirmed that independently of the category" to "In the FC".
Line 408: Change "DS" to "all types of DS".
Lines 407-8: By eye, it seems that not all of these are significantly different. To check this, I compared filo spines in Ctrl and Sham by estimating values from Figure 3G and doing a t-test. The p value (not assuming equal variances) was 0.1: nowhere close to 0.001. The fact that this huge error slipped past your editing suggests that there might be other statistical errors in the paper. Check all your statistics again throughout the whole paper.
Line 413: Add a close parenthesis.
Line 420: Add a close parenthesis.
Line 422: Change "remodeling dendritic alterations" to "dendritic remodeling".
Lines 422-4: This sentence is not contradicting the previous one. Rather it is pointing out that mushroom DS in particular show the effects of treatment. Since mushroom DS are mature, wouldn't this result imply that both interventions favor maturation of DS, rather than favoring both formation and maturation?
Figures 3F & 3G: Move the legend (Filo, Thin, Mushroom) from 3F to 3G. Consider changing the shapes in 3G to avoid confusion with the shapes in 3F, where the same shapes signify different things. Same for 3I & 3J.
Figure 3I: The difference between Sham and rTMS looks more significant than the difference between Sham and Ctrl because the rTMS data points are much more tightly clustered. I'm curious what the p values are.
Figure 3I: Change "Philo" to "Filo". Change the arrowhead to a circle.
Figure 3G and 3J: Maybe change the title from "Classification" to "Types of spines" or "Spine types" to be more descriptive.
Line 426: Change "on" to "of".
Line 427: Delete the comma.
Line 432: Change "about the distance" to "at particular distances (or up to 20 micrometers less)".
Line 433: Change "and FLX" to "or FLX". Change "reverse" to "reverses".
Lines 434-5: Change "the distance along the dendritic trees" to "Sham and the other groups at particular distances from the soma".
Line 436: Change "of the dendritic tree" to "from the soma".
Line 438: Change "(FC)," to "FC".
Lines 438-9: Change "dentate gyrus (DG)" to "DG".
Lines 439-441: You don't need to repeat the descriptions of all the groups. You already did that earlier in the caption for this figure. Just describe them as Ctrl, Sham, rTMS, and FLX, and indicate the photos for each.
Line 441: Remove "and G".
Line 445: Check the statistics in G and modify "Stress reduces in general all types of DS" accordingly.
Figure 3 caption: You forgot to mention I.
Line 447: Delete "(filopodia, filo; thin or mushroom) ".
Line 447: Change "decreases mainly," to "significantly decreases".
Line 448: Delete " type in the DG".
Line 448: Change "but rTMS" to "rTMS significantly".
Line 448: Change " type, while FLX" to ", and FLX significantly".
Line 449: Delete "the ".
Line 449: Delete "type".
Line 449: Change "symbols" to "symbol".
Line 450: Change "categories" to "DS types". "&&&" seems to be centered over mushrooms or cover all 3 types; state how you did the statistical tests.
Lines 450-1: Delete ", and in panel G". Did you mean that the following refers to panel G? If so, change ", and in panel G." to ". In panel G,".
Line 452: Delete ", and in panel J". Did you mean that the following refers to panel J? If so, change ", and in panel J." to ". In panel J,".
Figure 3 caption: The notation for statistical significance is complex and differs among lettered sections of the figure. Therefore, it may be clearer if you describe the notation after each lettered section of the Figure 3 caption rather than at the end of the whole Figure 3 caption.
Line 463: Delete "Thus, the quantification of the".
Line 464: Change ", D” to “ and D,”.
Line 467: Change “,” to “.”.
3.3: You analyze multiple dimensions of factors: marker, location, and group. Among these, you make many comparisons, which are lumped together without an apparent order in this big paragraph, making it hard for the reader to understand. You might organize it into smaller paragraphs to make your main points clearer. Another confusing writing convention is using “but” or “however” when it’s not clear what you’re contradicting. When I see “but”, I assume that it’s contradicting a previous observation, or your hypothesis.
Lines 470-1: You may delete this sentence because it’s not significant.
Line 471: Change "But, in the 5Hz rTMS group, there is a significant increase in" to “In the ML, compared to Sham,”.
Line 472: Change “compared to the Sham” to “increased in the rTMS group”.
Lines 472-3: Change “, and a similar but non-significant trend was found” to “ and showed a trend toward change”.
3.4:
Lines 479-492: These are the caption for Figure 4, so they should be integrated into the paragraph of lines 477-8.
Figure 4B, 4D, 4F, & 4H: I don’t understand the units. How did you get the area, such as 40000 square microns? Is that the area of the dashed square in the photo? It doesn’t seem to be, because the scale bar is 400 microns, so I guess the squares are about 250,000 square microns. Why are the areas different in each graph?
Line 479: Delete the comma.
Line 484: Change “nor” to “or”.
Lines 489-90: Change “CUMS decreases SYP immunoreactivity and rTMS reverses this effect in the ML” to “In the ML, rTMS increases SYP immunoreactivity compared to Sham”.
Lines 491-2: There’s no additional information in the colors, so delete this sentence and don’t use colors.
Line 493: You skipped section 3.4. I think this should be section 3.4.
Line 493: Change “and” to “or”.
Line 496: Delete “ initially,”.
Line 497: Delete “The “ from the beginning of the sentence.
Lines 500-503: Since these changes are not significant, maybe don’t even mention them. Just say something here as a placeholder to refer to Figure 5B.
Lines 503-5: With this word order, it implies that FLX caused higher immunoreactivity of H2BK16ac than FLX caused immunoreactivity of the other groups. But that’s not what you meant. Rewrite to clarify.
Line 525: Add “and” at the beginning of the line.
Line 525: Delete “Stress reduces the H2BK16ac immunoreactivity, but “ since it wasn’t significant.
Line 526: Change “it” to “H2BK16ac immunoreactivity”. After “Ctrl”, add “, Sham”.
Lines 536-7: Delete this sentence and the colors since they add no information.
Line 543: Change “Figure 6” and “Figure 7” to bold text.
Lines 546-549: Sentence fragment.
Line 549: Another sentence fragment.
Line 559: When you discuss global methylation, I assume you mean throughout all cells in the brain region. That seemed to be what you meant by "global" when discussing Figure 6E. But for Figure 6G, you imply that "global" means in neurons only.
Line 560: Shouldn't q be between zero and one?
Line 560: Change "and" to "or".
Line 560-1: Why do you say that FLX reversed the decrease in methylation, but in Figure 6G, FLX is not significantly different than Sham?
Line 561: p=0.050 is not significant.
Lines 563-566: I don't understand why you say the FLX didn't reverse the decrease of methylation in the first sentence, but you seem to say it did reverse it in the next sentence. I thought you're discussing Figure 6G in both sentences, but are you referring to different data in the first and second sentences?
Figure 6B&D: The units for the vertical axis label are confusing. Clearly indicate the units.
Figure 6E: The vertical axis units are mass, not concentration.
Figure 6G: Change "Immunofluorescency" to "Immunofluorescence".
Line 569: Change "citosine" to "cytosine".
Line 569: Add a comma after "cortex".
Line 570: You say "Scale bar" the first time and "scale bars" the second time. Later in the figure caption, you say "Scale bars". Be consistent throughout the paper.
Lines 568-583: Describe the subfigures in alphabetical order: A, B, C, D, E, F, and then G.
Line 575: Since you used past tense elsewhere, change "increases" to "increased" to be consistent.
Line 577: Add a comma before "and" to avoid confusion. Otherwise it sounds like you meant that stress decreased two things: 5mC and rTMS.
Line 579: For the first 3 photos in F, you list the color, then the stain, and then the name label for the photo. But for the 4th photo, you scramble the order. Be consistent.
Line 579: Insert "reduced" after "Stress".
Line 581: Change "represent" to "represents".
Lines 582-3: Remove the colors and this sentence because colors add no information here.
Line 587: Cite the reference using a number.
Line 591: Instead of giving the p values for differences from Ctrl, you should give the p values for differences from Sham because that's what you're talking about (not decreasing the immunoreactivity).
Line 592: Remove the extra close parenthesis.
Line 595: Only in FC, but not in DG.
Figure 7B&D: There's no need to stick an extra zero in the vertical axes labels. Just use 10 to the 4th power. Also, as in previous figures, clarify the units for these graphs.
Line 598: cytosine
Figure 7 caption: Describe in alphabetical order: A, B, C, D.
599: cytosine
605: represents
613: When you discuss the variance of the components here and later in the paper, 2 significant digits may be enough, and 3 seem plenty: no need for 4.
615: I think it reads easier if you start a new paragraph here, where you start to present the components.
617-8: I don't understand what you intended to say here. What do you mean by "for"? Re-write the sentence to clarify.
620: Change "," to ";".
623: Remove ".".
624-5: Yes, but component 2 also seems to separate Ctrl from the others. Is that right?
627-8: Sentence fragment.
635-6: Sentence fragment.
639: score plot
641: Add a "." after "treatments".
641: score plot
643: Change "In red show" to "Red indicates".
643: More percentage than what?
643: How did you set the cutoff for "more percentage"? It doesn't seem to be a certain percentage. Nor does it seem to be a certain number of parameters, like the top 5.
645: What do you mean by "n =3"? Is that the number of principal components?
647: For sections 2 & 3, the first text in each of those sections is a numbered subsection: 2.1 & 3.1. To be consistent, you may number the first paragraph of the Discussion as 4.1, and then renumber the subsequent subsections.
648: mechanisms
651-652: Saying "also" two times makes the conclusions seem redundant and tedious. Re-write to tie your 3 main points together in a more interesting way, for example by noting that your observations go from the big to the small: first at the behavioral level, then at the cellular level, and finally at the molecular level.
654-656: In general, introduce data in the Results, then interpret it in the Discussion. Don't introduce data for the first time in the Discussion. Thus, insert this observation into the Results, in an expanded form.
662: "most paradigms"? Do you mean "paradigms most commonly"?
662-3: "because reproduces"? Do you mean "because it reproduces"?
665-6: This list is a bit confusing because one might interpret it as meaning a deterioration in 3 things: CS, anhedonia, and despair behaviors. To avoid that possibility, you may change the order to put "deterioration in the CS" last.
669: Change "C57Bl/6" to "C57BL/6".
669: Delete "Also, ".
670: reversed
672: shown
672-3: Sentence fragment.
675: generated
677: Change "increased in the immobility behavior" to "immobility induced".
680: Omit the comma. Omit "may".
681: Change "been" to "with".
682: induce
687: probed
688: [29], [63].
689: Change "5. Hz" to " 5 Hz".
697: Why do you say "however"? It seems that this finding from previous studies is consistent with, rather than contradictory to, your findings.
698: decreased
699-700: Why do you say "In contrast"? That's the same as what you found: that thin type spines are most significantly affected by stress.
700: Change "are" to "were".
701: cell
702: Change ", CA1 region" to " or CA1 regions".
702-3: Add a comma before and after "as pointed out by Qio et al. 2016 [10]".
706: Add a comma before " confirming".
709: matrices
711-2: Sentence fragment.
714: Change ", shows" to " showed".
716: generated
718: Use some sort of punctuation to set off "1.14 Teslas", or precede it by "and ".
720: That's only a ~35% higher magnetic field. Is that really high intensity? Maybe change "high" to "somewhat higher". In fact, I looked at Table 1 of reference [67] and didn't see that 1.55 T generates the opposite effect. Rather, it looked like both 1.14 T and 1.55 T grew spines (though 1.55 T was less effective than 1.14 T).
730: Move the comma after "[69]".
734-5: "such SYP and RC3"?
740: Change "synapse dependent of” to “synapses dependent on”.
769-70: “functions that have also been found to be altered in patients with depression [85].” seems to be stuck on to the sentence without making sense.
771-2: This section subheading is too complicated.
773-4: There are two “and”s in this list. Just use “and” before the final item in a list.
775: “was found to decrease of”?
775-6: This contradicts your Figure 5B, which does not show significance.
778-9: “, H3 [19] and H5 [20] in the hippocampus and nucleus accumbens in models of social defeat stress.” seems to be stuck on to the sentence without making sense.
783: Use the standard spelling of this mouse line. Search for it online, find the correct spelling, and use that.
784-5: This contradicts your Figure 5H, where you wrote “Stress and the therapeutic interventions do not modify the H3K9me3 in the DG.”
787: “C57Bjl6”?
794: Add a comma after “[87]”.
797: Add “of “ before “histones”.
800: Change “vs” to “and”.
809: Change “GD” to “DG”.
809-11: I wouldn’t say that your data necessarily contrast with the Reszka study. That’s because you looked at the DG, while Reszka et al looked at blood. But if you want to compare the two studies, you could argue why we may expect that methylation would exhibit similar effects in brain as in blood. However, you just said in the previous paragraph that DNA methylation is particularly abundant in the brain. Presumably, that’s in contrast to the rest of the body.
811-2: I wrote the above comment before reading this sentence. It’s good you note this. However, it may be best not to even say there’s a contrast in the first place because saying there’s a contrast may mislead and confuse readers, and waste their time.
826: Add a comma before “ who”.
832: Change “the dendritic spines” to “dendritic spine”.
843: Change “GD” to “DG”.
844-8: There are two sentences stuck together are a grammar error, it’s a comma splice.
856-7: Change “, in the case of rTMS, they do” to “ rTMS does”.
859: Change “GD” to “DG”.
877: facilitates
897-8: “Its use in the biomedical area (psychiatric, neurodegenerative, motor diseases, etc.) is enhanced and expanded.” Explain. How, when, why? Do you mean by the results in your paper? Or do you mean in general in recent years?
906-911: There are a bunch of missing periods on some initials. There’s an extra quotation mark.
913: ..
916: Missing period
928: World Health Organization COVID-19 Pandemic... implies that it is the WHO’s pandemic. Add a period after the author and before the title.
939: Add the journal name and volume number.
954-962: Is reference 14 the same reference as 15?
1011-3: Effects. Add the journal name and volume number.
1014-5: Incomplete list of authors. Wrong journal name. Wrong volume number. Wrong page number.
1020-4: Reference 39 and 40 refer to the same paper.
1057-61: References 56 and 57 refer to the same paper.
1065-8: Remove the letters and numbers after each author’s name. Use initials for authors, just as for other references.
1069-70: Check the authors’ names.
1075-7: Check the authors’ names.
1084: Add the journal name.
1086: Add the journal name.
1092: Add the journal name.
1144: DNA
1155-6: Is an author actually named C.H. Disease?!
Supplementary: Change “performed” to “constructed”.
Supplementary: Change “groups” to “groups,”.
Supplementary: Change “In general, it can be observed that there are different patterns of correlation in each of the experimental groups,“ to “In general, it can be observed that there are different patterns of correlation in each of the experimental groups;“.
Supplementary: Change “act modifying” to “act by modifying”.
Supplementary: Delete “it observed that “.
Supplementary: Change “spines with” to “spines and”.
Supplementary: Change “p= 0.49” to “p= 0.049”.
Supplementary: Change “Also exists” to “There is”.
Supplementary: Change “p= 0.49” to “p= 0.049”.
Supplementary: Change “Also exists” to “There is”.
Supplementary: This sentence is complicated, so you may split it into two: “Also exists a negative correlation of the DNA methylation of mature neurons and the trimethylation of histone H3 in the DG with the mature and intermediate dendritic spines in the Hp (r= -0.999, p= 0.030, r= -0.998, p= 0.043 respectively).”
See above
Author Response
REVIEWER 3: Repetitive transcranial magnetic stimulation (rTMS) is a valuable emerging option for treating depression. It is important to explore mechanisms by which rTMS works. This paper is a worthwhile examination of the effects of rTMS on neuronal structures and epigenetics in mice.
The writing starts off clear. However, it later displays many errors. There are a huge number of careless mistakes throughout most of the paper, and I tried to list them below.
We thank the observations of this reviewer. In the new version of our manuscript, we corrected the mistakes indicated.
Line 59: Change "intermediated" to "intermediate".
A: Corrected
Line 62: Change "desoxyribonucleic" to "deoxyribonucleic".
A: Corrected
Line 74: Change "DNA." to "DNA".
A: Corrected
Line 75: Change "," to ".".
A: Corrected
Line 84: Change "generate different alterations in the brain ranging from the" to "changes brain", or something like that.
A: Corrected
Line 86: Delete "the" before BDNF, unless you're discussing a particular subset of BDNF expression (in which case you should specify which BDNF expression is increased).
A: Corrected
Line 88: Move the comma to before "and".
A: Corrected
Line 89: Change "the long-term potentiation" to "long-term potentiation".
A: Corrected
Line 105: Change "Balb/C" to "BALB/c" here and throughout the paper.
A: Corrected
Line 109: Change "along" to "during the", or something like that.
A: Corrected
Lines 123-125: In this section, explain more clearly how you applied CUMS. You gave 1-3 stressors per day, but how did you choose the number of stressors to apply each day? Was that random? Once the number was chosen, how did you choose a stressor from Table 1? Was that also random? Some stressors last 12-24 hours, thus they can't be applied 2 or 3 times a day. How did you deal with that? Or did you choose one stressor each day and then apply that same stressor all day?
- How did you choose the number of stressors to apply each day?
A: We applied three stressors per day, with occasional days (one or two per week) where one or two stressors were applied to introduce variability into the stress protocol.
- Was that random?
A: The choice of the number of stressors per day was not entirely random, as we aimed to avoid repeating the same number of stressors on consecutive days. It was done to create an unpredictable stress exposure.
Once the number was chosen, how did you choose a stressor from Table 1?
- A: The same stressor was not repeated on consecutive days at the same time, with the type of stressor varying each day.
- The "continuous light" and "strobe light" stressors were applied during the dark phase of the mice's light-dark cycle to generate a stress response by altering their photoperiod. The "strobe light" stressor, which lasts for 4 hours, was applied at different times within the dark phase.
- Three stressors for 12 or 24 hours could not be applied on the same day.
- Only the "food deprivation" stressor was always applied on Fridays, as this is a prerequisite for the sucrose preference test [9]. The sucrose preference test was conducted on Saturdays at the same time to prevent changes in timing from affecting measurements in the test.
Was that also random?
- A: The selection of stressors applied per day was not entirely random, as the types of stressors chosen depended on avoiding the repetition of the same stressor on consecutive days. It was done to prevent the formation of patterns and to make the protocol unpredictable.
- Some stressors last 12-24 hours, thus they can't be applied 2 or 3 times a day. How did you deal with that?
A: No more than two stressors with a duration of 12 to 24 hours were applied on the same day to prevent the mice from being constantly stressed. A maximum of two stressors with a duration of 12 or 24 hours could be applied simultaneously on the same day. For example, the "continuous light" stressor only becomes stressful in the last hours of its application, disrupting the mice's 12-hour circadian cycles. During the first 12 hours of continuous light, applying the "white noise" or "wet bed" stressors was possible.
Line 129: Change "USA)" to "USA". Change "wide" to "width".
A: Corrected
Line 130: Change "amplitude/magnetic field strength" to "change in magnetic field strength per time".
A: Corrected
Line 133-4: Change "10%, increasing every 2%" to "10% of maximum, increasing by 2% of maximum".
A: Corrected
Line 140: Instead of "0" to indicate degrees, use the actual degree symbol.
A: Corrected
Lines 155-158: You listed the groups twice.
A: Corrected
Line 156: Change "Kg" to "kg" throughout the paper.
A: Corrected
Line 162: Change "magnetic field strength" to "maximum rate of change in magnetic field strength".
A: Corrected
Lines 186-192: Why did you restrict food and water after the SPT? Why did you change the protocol from that in reference 42?
A: We apologize for this mistake. In the new version of our manuscript, we correct it as follows: The SPT consisted of depriving mice of food and water for 18 hours. Then evaluating the consumption of a sucrose solution (1%) or natural water in an individual box (19.05x29.21x12.7 cm)
Line 189: Change "Week basal" to "basal week".
A: Corrected
Line 199: Is the phrase in parentheses an example? If so, indicate it.
A: We correct it as follows: The criteria for considering immobility in the mice were that 90% of their whole body remained motionless for a minimum period of 2.5 seconds.
Line 203: What do you mean by "half": every other brain, or one hemisphere of each brain? I assume you mean the former. If so, one way to clarify this would be to say, "about half". Anyway, "about half" is more correct than just "half" because some groups had odd numbers of mice (for example, n=7).
A: The text refers to the hemispheres of the brain. To clarify, we have revised the sentence: After euthanizing the mice, their brains were extracted. Four out of seven or ten brains were designated for histological processing. The left hemispheres were used for Golgi-Cox impregnation, while the right hemispheres were designated for immunohistochemistry and immunofluorescence (Figure 2). Additionally, three brains from each group were used to dissect the dentate gyrus (DG) (Figure 2).
Line 209: Spell out what 5mC is the first time you mention it.
A: We spelled out 5mC.
Line 230: What do you mean by "rate"?
A: The term "rate length/width" refers to the value resulting from the division of the length by the width of a dendritic spine size.
Line 233 and throughout the paper: Delete "the" before "Sholl".
A: Corrected
Line 235-238: It's confusing that you describe killing both here and section 2.7. Consolidate these and only describe killing once. You might add a small flow chart indicating the steps and numbers of mice: 7-10 mice per experimental group, which were all killed and brains removed, of which about half were used to extract DNA, one was Golgi Cox stained, and the remainder were used for immunohistochemistry. It took me quite a while to piece together the above flowchart from the text.
A: It is true. We corrected it, and in the new version of our manuscript, we only included information about the sacrifice in section 2.7. The information included was: Once mice were euthanized, the brains were extracted, and 4 from 7 or 10 brains were destined for histological processing. Thus, the left hemispheres were used for Golgi Cox impregnation, and the right hemispheres were destined for immunohistochemistry and immunofluorescence (Figure 2). Also, 3 brains from each group were used to dissect the DG (Figure 2).
Line 258: Change "Merk" to "Merck". I couldn't find "Newmont medium" anywhere on the Merck SigmaAldrich website or even anywhere on the internet. You may list the catalog number in the paper.
A: Corrected
Line 260: Delete "Images of " because you say "were imaged" later.
A: Corrected
Line 262: Add a period at the end of the sentence (after "v. 3.4.0").
A: Corrected
Lines 266-271: What do you mean by "quadrants"? Is the microscope field divided into 4 parts? Why quadrants instead of whole fields?
A: The correct term is "squares," not "quadrants," since 200x200 and 50x50 micrometer squares were used to perform the optical density measurements of the markers. The field was not divided into four parts. The term is replaced in the text.
Line 273: You skipped section 2.10.
A: Corrected
Line 273: Add a period at the end, to be consistent with your other subheadings.
A: Corrected
Sections 2.9-2.11: The section headings are confusing. You may rearrange and retitle these sections, perhaps so that there is one heading on immunohistochemistry, with subheadings for tissue processing for immunohistochemistry, synapse proteins, and 5mC.
A: Corrected
Line 293: Change the period to a comma.
A: Corrected
Line 297: Change "Glomax discover, Promega." to "GloMax Discover, Promega,".
A: Corrected
Line 302: Referring to data as ordered in distributions doesn't seem quite correct. Maybe you're referring to the ordering of data in nonparametric tests. But I think it makes more sense to replace "in which our data were ordered" with "of our data".
A: Corrected
Lines 311-6: I have no experience with PCA, so I can't evaluate this.
A: Understood
Line 318: Change "and" to "or".
A: Corrected
Lines 320-2: This sentence has some redundancies. You may change it to something like: "The coat deteriorated in chronically stressed mice compared to control mice (p<0.001)."
A: Corrected
Line 323: The 4th week doesn't show the worst CS. It looks like the 8th week is worse. You may change "the highest" to "a plateau in the" and delete the subsequent sentence.
A: Corrected
Line 329 and later: You don't need 5 significant digits in the F values. 3 seems plenty. By analogy, if you asked me my age, you'd think I was weird if I said I'm 58.284 years old. You probably only want to hear 2 significant digits for my age. Throughout the paper, reduce the number of significant digits to something reasonable.
A: Corrected
Line 331 and later: Change "X" to "×".
A: Corrected
Line 331: Add a close parenthesis.
A: Corrected
Line 335 and later: "Revert" is an intransitive verb (https://www.merriam-webster.com/dictionary/revert). Change "reverted" to "reversed".
A: Corrected
Line 335: It's surprising that rTMS and FLX are not significantly different than controls in week 6. Looking at the error bars in the graph, they look different. But maybe looks are deceiving.
Q: There are no significant differences between the control group compared to the rTMS group (p= 0.087) and compared to the FLX group (p= 1.00) at week 6 (Two-way repeated measures ANOVA. Bonferroni test post hoc).
Line 336: Move "mice" before "without".
A: Corrected
Line 339: Add a close parenthesis.
A: Corrected
Line 345: Change "number" to "increase in number".
A: Corrected
Lines 345-6: Why do you show 2 p-values?
Q: These are the "p" values of the number of episodes and the immobility time in the forced swim test. It is corrected and specified in the text to which parameter each value of "p" corresponds.
Line 347: Why do you show 2 p-values?
Q: The first (p= 0.022) is the p-value of the comparison between the FLX and the Sham groups in the number of immobility episodes. The second (p= 0.002) is the p-value of the one-way ANOVA of the number of immobility episodes of the forced swim test. The p-values in the text are corrected, and parentheses are added to clarify what each value corresponds to.
Figure 2D: In the vertical axis label, delete "(s)".
A: Corrected
Line 352: Change “Control" to "control".
A: Corrected
Line 355: Change "and" to "or".
A: Corrected
Line 355 and throughout the paper: Put the adjective before the noun. Change “ANOVA two-way repeated measurement” to “Two-way repeated measures ANOVA”. From this point onward in the paper, you often reversed the order of the name of ANOVA tests. For example: “ANOVA one-way”. It should be “one-way ANOVA”. Find these throughout the paper and correct them.
A: We apologize for those mistakes. We corrected them.
Figure 2: If you want to show significance with different symbols in each part of the figure, you should state the explanation for each symbol you use in the legend for each part of the figure (A, B, C, and D). Alternatively, you can devise symbols that work for all parts of the figure; then you need to explain them only once in the legend: at the end of the first part of the legend, before the specific legend for part A.
A: Corrected
Lines 355-6: Why do you say "ANOVA two-way repeated measurement. Bonferroni test post hoc." without making any quantitative statement about statistical results in Figure 2A?
Q: In the figure caption after the description of each panel, only the statistical test used is added because the statistical values are described in the results.
Line 356: Delete ",".
A: Corrected
Line 359: Change "along the time exposure to the CUMS versus the Ctrl group" to "between the CUMS and the Ctrl groups".
A: Corrected
Lines 360-1: Explain what levels of significance && and &&& indicate.
A: Corrected
Line 366: Stress does not significantly increase the immobility time. It looks like a trend, though.
A: Corrected
Lines 366-7: FLX does not significantly reverse the effect. It may be a trend.
A: Corrected
Line 370: Change "Line" to "Error".
A: Corrected
Line 376: Change "dendrite trees in" to "dendritic trees of".
A: Corrected
Line 380: Add "the" before "control".
A: Corrected
Line 381: Change "from 20 to 140 micrometers related to the soma" to "at distances of 20 to 140 micrometers from the soma".
A: Corrected
Lines 381-2: Are these p values correct? The figure caption says p<0.001.
Q: The p values were verified and corrected by specifying them in the text and figure caption.
Line 385: Add a close parenthesis.
A: Corrected
Line 386: It's not trees that are impregnated with dendrites, so change "dendrite-impregnated trees in" to "dendritic trees of". I think it's not necessary to state again here that you're looking only at impregnated cells; you've already described this in the Methods.
A: Corrected
Line 392: Change "from 60 to 180 micrometers related to the soma" to "at distances of 60 to 180 micrometers from the soma".
A: Corrected
Line 395: Add a close parenthesis.
A: Corrected
Line 396: Change "concerning the distance" to "at different distances".
A: Corrected
Lines 400-3: Define the q values you use. Usually, q values are between 0 and 1. Explain the statistical analysis you used, and how you examined significance.
A: A one-way ANOVA on ranks was performed with a Student-Newman-Keuls (SNK) post hoc test. The q value in this post hoc test is used to evaluate the difference between group means. It is calculated by subtracting the means of the groups being compared and dividing the result by the average standard error of the groups. For a significant difference between the means of the groups being compared, the value of q must be greater than the critical t-value [10]. For a significance level of 5% with 3 degrees of freedom and a total sample size 18, the critical t-value is 3.1824 [11].
Line 407: Change "The FC confirmed that independently of the category" to "In the FC".
A: Corrected
Line 408: Change "DS" to "all types of DS".
A: Corrected
Lines 407-8: By eye, it seems that not all of these are significantly different. To check this, I compared filo spines in Ctrl and Sham by estimating values from Figure 3G and doing a t-test. The p value (not assuming equal variances) was 0.1: nowhere close to 0.001. The fact that this huge error slipped past your editing suggests that there might be other statistical errors in the paper. Check all your statistics again throughout the whole paper.
A: We have rechecked the statistics throughout the text as suggested and made the necessary corrections. In the specific case of Figure 3G, as you pointed out, there is no significant difference in the number of phylum spines between the Ctrl and Sham groups (p=0.331). The symbol “***" represents a significant difference with a p-value < 0.001 between the Ctrl and Sham groups without considering the morphology of dendritic spines (indicated by a line above the types of dendritic spines for these groups without specifying any particular type). It represents the main effects of the treatment factor. In the caption for Figure 3 (lines 464-465), this is specified: "in panel G. (*) symbols represent differences between groups independently of the DS category (**: p ≤ 0.05, ***: p ≤ 0.001)”.
Line 413: Add a close parenthesis.
A: Corrected
Line 420: Add a close parenthesis.
A: Corrected
Line 422: Change "remodeling dendritic alterations" to "dendritic remodeling".
A: Corrected
Lines 422-4: This sentence is not contradicting the previous one. Rather it is pointing out that mushroom DS in particular show the effects of treatment. Since mushroom DS are mature, wouldn't this result imply that both interventions favor maturation of DS, rather than favoring both formation and maturation?
A: The sentence is rewritten so that it is not interpreted as a contradiction with the previous sentence, and the correction is made that the increase in the number of mushroom-like spines suggests that both 5Hz rTMS and FLX favor the maturation process. On the other hand, it should be noted that the increase in the density of dendritic spines in these treatments, concerning the Sham group, could be indicating that both 5Hz rTMS and FLX also promote the formation of new spines, as well as their maturation.
Figures 3F & 3G: Move the legend (Filo, Thin, Mushroom) from 3F to 3G. Consider changing the shapes in 3G to avoid confusion with the shapes in 3F, where the same shapes signify different things. Same for 3I & 3J.
A: Corrected
Figure 3I: The difference between Sham and rTMS looks more significant than the difference between Sham and Ctrl because the rTMS data points are much more tightly clustered. I'm curious what the p values are.
A: To analyze the total number of dendritic spines in the hippocampal dentate gyrus (Figure 3I), we performed a one-way ANOVA on ranks with a Student-Newman-Keuls post hoc test. In this post hoc test, the significance of group comparisons is based on the q-value. If the q-value is greater than the critical t-value (3.1824), a p-value < 0.05 is considered significant, regardless of the degree of significance. It is why all significant differences are marked with the symbol “*,” and all comparisons can be interpreted as having the same degree of significance.
Figure 3I: Change "Philo" to "Filo". Change the arrowhead to a circle.
A: Corrected
Figure 3G and 3J: Maybe change the title from "Classification" to "Types of spines" or "Spine types" to be more descriptive.
A: Corrected
Line 426: Change "on" to "of".
A: Corrected
Line 427: Delete the comma.
A: Corrected
Line 432: Change "about the distance" to "at particular distances (or up to 20 micrometers less)".
A: Corrected
Line 433: Change "and FLX" to "or FLX". Change "reverse" to "reverses".
A: Corrected
Lines 434-5: Change "the distance along the dendritic trees" to "Sham and the other groups at particular distances from the soma".
A: Corrected
Line 436: Change "of the dendritic tree" to "from the soma".
A: Corrected
Line 438: Change "(FC)," to "FC".
A: Corrected
Lines 438-9: Change "dentate gyrus (DG)" to "DG".
A: Corrected
Lines 439-441: You don't need to repeat the descriptions of all the groups. You already did that earlier in the caption for this figure. Just describe them as Ctrl, Sham, rTMS, and FLX, and indicate the photos for each.
A: Corrected
Line 441: Remove "and G".
A: Corrected
Line 445: Check the statistics in G and modify "Stress reduces in general all types of DS" accordingly.
A: Corrected
Figure 3 caption: You forgot to mention I.
A: Corrected
Line 447: Delete "(filopodia, filo; thin or mushroom) ".
A: Corrected
Line 447 (454): Change "decreases mainly," to "significantly decreases".
A: Corrected
Line 448 (454): Delete " type in the DG".
A: Corrected
Line 448 (454): Change "but rTMS" to "rTMS significantly".
A: Corrected
Line 448 (454): Change " type, while FLX" to ", and FLX significantly".
A: Corrected
Line 449: Delete "the ".
A: Corrected
Line 449: Delete "type".
A: Corrected
Line 449: Change "symbols" to "symbol".
A: Corrected
Line 450: Change "categories" to "DS types". "&&&" seems to be centered over mushrooms or cover all 3 types; state how you did the statistical tests.
A: Corrected. To analyze the differences between groups by dendritic spine type in the frontal cortex (Figure 3G) and dentate gyrus (Figure 3J), we performed a two-way ANOVA with a Bonferroni post hoc test. Differences between the main factors and their interaction were compared. In panels G and J, the symbol (&) represents a significant difference between dendritic spine types within the same group (&&&: p ≤ 0.001). The statistical values of the two-way ANOVA are shown in the table below, along with comparisons of the “Morphology” (dendritic spine type) factor by group and their corresponding p-values:
DS type in FC (Figure3G)
Source of Variation DF SS MS F P
Treatments 3 15.191 5.064 10.202 <0.001
Morphology 2 87.804 43.902 88.449 <0.001
Treatments x Morphology 6 6.331 1.055 2.126 0.074
Residual 36 17.869 0.496
Total 47 127.195 2.706
All Pairwise Multiple Comparison Procedures (Bonferroni t-test):
Comparisons for factor: Treatments
Comparison Diff of Means t P P<0.050
Ctrl vs. Sham 1.575 5.476 <0.001 Yes
Ctrl vs. Flx 0.829 2.883 0.040 Yes
Ctrl vs. Rtms 0.621 2.159 0.226 No
Rtms vs. Sham 0.954 3.317 0.013 Yes
Rtms vs. Flx 0.208 0.724 1.000 No
Flx vs. Sham 0.746 2.593 0.082 No
Comparisons for factor: Morphology within Ctrl
Comparison Diff of Means t P P<0.05
Mush vs. Filo 3.637 7.302 <0.001 Yes
Mush vs. Thin 3.575 7.176 <0.001 Yes
Thin vs. Filo 0.0625 0.125 1.000 No
Comparisons for factor: Morphology within Sham
Comparison Diff of Means t P P<0.05
Mush vs. Filo 1.700 3.412 0.005 Yes
Mush vs. Thin 1.462 2.936 0.017 Yes
Thin vs. Filo 0.238 0.477 1.000 No
Comparisons for factor: Morphology within Rtms
Comparison Diff of Means t P P<0.05
Mush vs. Filo 3.125 6.273 <0.001 Yes
Mush vs. Thin 3.025 6.072 <0.001 Yes
Thin vs. Filo 0.1000 0.201 1.000 No
Comparisons for factor: Morphology within Flx
Comparison Diff of Means t P P<0.05
Mush vs. Filo 3.325 6.674 <0.001 Yes
Mush vs. Thin 3.075 6.173 <0.001 Yes
Thin vs. Filo 0.250 0.502 1.000 No
DS type in DG (Figure3J)
Normality Test (Shapiro-Wilk) Passed (P = 0.336)
Equal Variance Test: Passed (P = 0.835)
Source of Variation DF SS MS F P
Group 3 18.941 6.314 12.800 <0.001
Morphology 2 55.128 27.564 55.883 <0.001
Group x Morphology 6 14.057 2.343 4.750 <0.001
Residual 41 20.223 0.493
Total 52 109.037 2.097
Comparisons for factor: Treatment
Comparison Diff of Means t P P<0.050
Flx vs. Sham 1.567 5.464 <0.001 Yes
Flx vs. Ctrl 0.216 0.800 1.000 No
Flx vs. Rtms 0.204 0.712 1.000 Do Not Test
Rtms vs. Sham 1.363 4.752 <0.001 Yes
Rtms vs. Ctrl 0.0115 0.0428 1.000 Do Not Test
Ctrl vs. Sham 1.351 5.013 <0.001 Yes
Comparisons for factor: Morphology within Ctrl
Comparison Diff of Means t P P<0.05
Mush vs. Filo 2.021 4.699 <0.001 Yes
Mush vs. Thin 0.194 0.413 1.000 No
Thin vs. Filo 1.827 4.562 <0.001 Yes
Comparisons for factor: Morphology within Sham
Comparison Diff of Means t P P<0.05
Mush vs. Filo 2.313 4.657 <0.001 Yes
Mush vs. Thin 1.925 3.876 0.001 Yes
Thin vs. Filo 0.387 0.780 1.000 No
Comparisons for factor: Morphology within Rtms
Comparison Diff of Means t P P<0.05
Mush vs. Thin 3.212 6.469 <0.001 Yes
Mush vs. Filo 2.262 4.556 <0.001 Yes
Filo vs. Thin 0.950 1.913 0.188 No
Comparisons for factor: Morphology within Flx
Comparison Diff of Means t P P<0.05
Mush vs. Filo 3.088 6.217 <0.001 Yes
Mush vs. Thin 2.525 5.084 <0.001 Yes
Thin vs. Filo 0.563 1.133 0.792 No
Lines 450-1: Delete ", and in panel G". Did you mean that the following refers to panel G? If so, change ", and in panel G." to ". In panel G,".
A: Corrected
Line 452: Delete ", and in panel J". Did you mean that the following refers to panel J? If so, change ", and in panel J." to ". In panel J,".
A: Corrected
Figure 3 caption: The notation for statistical significance is complex and differs among lettered sections of the figure. Therefore, it may be clearer if you describe the notation after each lettered section of the Figure 3 caption rather than at the end of the whole Figure 3 caption.
A: Corrected
Line 463 (475): Delete "Thus, the quantification of the".
A: Corrected
Line 464: Change ", D” to “ and D,”.
A: Corrected
Line 467 (478): Change “,” to “.”.
3.3: You analyze multiple dimensions of factors: marker, location, and group. Among these, you make many comparisons, which are lumped together without an apparent order in this big paragraph, making it hard for the reader to understand. You might organize it into smaller paragraphs to make your main points clearer. Another confusing writing convention is using “but” or “however” when it’s not clear what you’re contradicting. When I see “but”, I assume that it’s contradicting a previous observation, or your hypothesis.
A: We apologize for the lack of clarity. In the new version of our manuscript, we reordered this paragraph as follows:
“RC3 in the FC (Figure 5A, B) and ML in the DG (Figure 5C, D) did not show differences among the groups (FC: One-way on ranks ANOVA, H = 4, df = 3, p = 0.210, ML: One-way ANOVA, F3,15 = 2, p = 0.106).
In the case of SYP in the FC (Figure 5E, F), it increased in mice treated with the 5Hz rTMS (p = 0.013) or FLX (p < 0.001) compared to the stressed mice. In the case of FLX, this treatment was significantly different compared with the Ctrl (p < 0.001) and rTMS (p = 0.020) groups (One-way ANOVA, F3,15 = 23, p < 0.001). In the ML in the DG (Figure 5G, H) compared to Sham, SYP immunoreactivity increased in the rTMS group (p = 0.048) and showed a trend toward chance in the FLX group (Figure 5H) (p = 0.077; One-way ANOVA, F3,15 = 4, p = 0.026). These results suggest that both treatments, 5Hz rTMS and FLX, favored the increased expression of SYP, a protein located at the presynaptic vesicles [52].”
Lines 470-1: You may delete this sentence because it’s not significant.
A: Corrected
Line 471: Change "But, in the 5Hz rTMS group, there is a significant increase in" to “In the ML, compared to Sham,”.
A: Corrected
Line 472: Change “compared to the Sham” to “increased in the rTMS group”.
A: Corrected
Lines 472-3 (484): Change “, and a similar but non-significant trend was found” to “ and showed a trend toward change”.
A: Corrected
3.4:
Lines 479-492: These are the caption for Figure 4, so they should be integrated into the paragraph of lines 477-8.
A: Corrected
Figure 4B, 4D, 4F, & 4H: I don’t understand the units. How did you get the area, such as 40000 square microns? Is that the area of the dashed square in the photo? It doesn’t seem to be, because the scale bar is 400 microns, so I guess the squares are about 250,000 square microns. Why are the areas different in each graph?
A: Immunoreactivity can indicate the levels of a ligand (in our work: RC3, SYP, H2BK16ac, H3K9me3, 5mC, and h5mC) present in a tissue using the immunohistochemistry technique. This parameter is measured as optical density in arbitrary units, calculated based on the number of pixels found in a unit area. Higher immunoreactivity of an antigen-antibody junction contains a higher number of pixels per area, resulting in a higher optical density measurement. The dotted boxes in the upper photos of panels A, C, E, and G represent the areas where the 40x photos were taken and are located at the bottom of the panels. To calculate the area used for frontal cortex (FC) measurements (40,000 micrometers), we created quadrants of 200x200 micrometers (200x200=40,000). To obtain the area used for dentate gyrus (DG) measurements (2,500 micrometers), quadrants of 50x50 micrometers (50x50=2,500) were created. This process was performed in ImageJ software using the “rectangle” tool after calibrating the photo. The reason for using different areas to measure optical density in the FC and DG is due to the difference in the size of these areas. The average thickness of layers II/III of the cortex in mice is 200 micrometers, while that of the DG is 50 micrometers [12].
Line 479: Delete the comma.
A: Corrected
Line 484 (496): Change “nor” to “or”.
A: Corrected
Lines 489-90 (499): Change “CUMS decreases SYP immunoreactivity and rTMS reverses this effect in the ML” to “In the ML, rTMS increases SYP immunoreactivity compared to Sham”.
A: Corrected
Lines 491-2: There’s no additional information in the colors, so delete this sentence and don’t use colors.
A: Corrected
Line 493: You skipped section 3.4. I think this should be section 3.4.
A: Corrected
Line 493: Change “and” to “or”.
A: Corrected
Line 496 (506): Delete “ initially,”.
A: Corrected
Line 497: Delete “The “ from the beginning of the sentence.
A: Corrected
Lines 500-503: Since these changes are not significant, maybe don’t even mention them. Just say something here as a placeholder to refer to Figure 5B.
A: Corrected
Lines 503-5: With this word order, it implies that FLX caused higher immunoreactivity of H2BK16ac than FLX caused immunoreactivity of the other groups. But that’s not what you meant. Rewrite to clarify.
A: Corrected
Line 525: Add “and” at the beginning of the line.
A: Corrected
Line 525 (531): Delete “Stress reduces the H2BK16ac immunoreactivity, but “ since it wasn’t significant.
A: Corrected
Line 526: Change “it” to “H2BK16ac immunoreactivity”. After “Ctrl”, add “, Sham”.
A: Corrected
Lines 536-7: Delete this sentence and the colors since they add no information.
A: Corrected
Line 543 (549): Change “Figure 6” and “Figure 7” to bold text.
A: Corrected
Lines 546-549: Sentence fragment.
A: We corrected it as follows: However, in the DG (Figure 7C, D), rTMS increased the immunoreactivity for this marker compared to the control group (p = 0.015; One-way ANOVA, F3,14 = 5, p = 0.014).
Line 549: Another sentence fragment.
A: We corrected it as follows: To confirm the increased 5mC immunoreactivity in the DG, we quantified the 5mC concentration by ELISA (Figure 7E). The CUMS decreased the concentration of 5mC compared to the control group (p = 0.028). The effect caused by the CUMS was reversed by rTMS (p = 0.015) but not by FLX (p = 1.00; One-way ANOVA, F3,11 = 15, p = 0.001).
Line 559: When you discuss global methylation, I assume you mean throughout all cells in the brain region. That seemed to be what you meant by "global" when discussing Figure 6E. But for Figure 6G, you imply that "global" means in neurons only.
A: Your observation is correct; "global methylation" refers to methylation occurring in different cell types. We correct the text:
“The 5mC expression in NeuN cells revealed that the CUMS significantly decreased DNA methylation compared to the control group (q = 3.683). Still, applying rTMS or FLX reverses the decrement (q = 4.006 and q = 3.703, respectively. One-way ANOVA on ranks, H = 8, df = 3, p = 0.050). These data confirm that the rTMS increased the global DNA methylation in the DG. Interestingly, mature neurons increase DNA methylation. The treatment with FLX did not reverse the decrement of global DNA methylation, but it did reverse the decrease in the methylation of mature neurons in the DG of the hippocampus. This may suggest that at the level of DNA methylation, rTMS may act on other types of cells residing in the hippocampus, whereas FLX seems to act more on mature neurons of DG. “
Line 560: Shouldn't q be between zero and one?
A: The "q" value of the student-Newman-Keuls post hoc test depends on the difference between the means and the standard error of the groups to be compared. Therefore, its value may not be between 1 and 0 [10].
Line 560: Change "and" to "or".
A: Corrected
Line 560-1: Why do you say that FLX reversed the decrease in methylation, but in Figure 6G, FLX is not significantly different than Sham?
A: Figure 6G is corrected since there is a significant difference between the FLX and Sham groups (3.703).
Line 561: p=0.050 is not significant.
A: The significance value established at the beginning of our analysis was p ≤ 0.05. A p-value of 0.05 is the threshold for accepting or rejecting the null hypothesis, and opinions are divided on whether it should be considered significant. It is generally accepted that p-values less than 0.05 provide sufficient evidence to reject the null hypothesis and suggest a biological effect between the groups analyzed. Our study used a significance value of p ≤ 0.05 due to the sample size used in our experiments.
Lines 563-566: I don't understand why you say the FLX didn't reverse the decrease of methylation in the first sentence, but you seem to say it did reverse it in the next sentence. I thought you're discussing Figure 6G in both sentences, but are you referring to different data in the first and second sentences?
A: In these lines, we first discuss the results of global DNA methylation and then the results of DNA methylation in mature neurons of the hippocampal dentate gyrus (DG). We have revised the text to make the idea more straightforward: Interestingly, mature neurons increase DNA methylation. While FLX treatment did not reverse the decrease in global DNA methylation, it did reverse the decrease in methylation of mature neurons in the hippocampal DG. This may suggest that rTMS may act on other types of cells residing in the hippocampus, whereas FLX seems to have a more significant effect on mature neurons of the DG.
Figure 6B&D: The units for the vertical axis label are confusing. Clearly indicate the units.
A: Corrected
Figure 6E: The vertical axis units are mass, not concentration.
Q: Units are corrected to “ng/µl”.
Figure 6G: Change "Immunofluorescency" to "Immunofluorescence".
A: Corrected
Line 569: Change "citosine" to "cytosine".
A: Corrected
Line 569: Add a comma after "cortex".
A: Corrected
Line 570: You say "Scale bar" the first time and "scale bars" the second time. Later in the figure caption, you say "Scale bars". Be consistent throughout the paper.
A: Corrected
Lines 568-583: Describe the subfigures in alphabetical order: A, B, C, D, E, F, and then G.
A: Corrected
Line 575: Since you used past tense elsewhere, change "increases" to "increased" to be consistent.
A: Corrected
Line 577: Add a comma before "and" to avoid confusion. Otherwise, it sounds like you meant that stress decreased two things: 5mC and rTMS.
A: Corrected
Line 579: For the first 3 photos in F, you list the color, then the stain, and then the name label for the photo. But for the 4th photo, you scramble the order. Be consistent.
A: Corrected
Line 579: Insert "reduced" after "Stress".
A: Corrected
Line 581: Change "represent" to "represents".
A: Corrected
Lines 582-3: Remove the colors and this sentence because colors add no information here.
A: Corrected
Line 587: Cite the reference using a number.
A: Corrected
Line 591: Instead of giving the p values for differences from Ctrl, you should give the p values for differences from Sham because that's what you're talking about (not decreasing the immunoreactivity).
A: We corrected it as follows: “At the FC (Figure 8A, B), the CUMS exposed groups significantly increased the 5hmC immunoreactivity compared with the control (p < 0.001). Interestingly, mice treated with rTMS or FLX do not show significant changes concerning the CUMS group (rTMS: p < 0.001; and FLX: p < 0.001), (One-way ANOVA, F3,15 = 20, p < 0.001).”
Line 592: Remove the extra close parenthesis.
A: Corrected
Line 595: Only in FC, but not in DG.
A: Corrected
Figure 7B&D: There's no need to stick an extra zero in the vertical axis’s labels. Just use 10 to the 4th power. Also, as in previous figures, clarify the units for these graphs.
A: Corrected
Line 598: cytosine
A: Corrected
Figure 7 caption: Describe in alphabetical order: A, B, C, D.
A: Corrected
599: cytosine
A: Corrected
605: represents
A: Corrected
613: When you discuss the variance of the components here and later in the paper, 2 significant digits may be enough, and 3 seem plenty: no need for 4.
A: Corrected
615: I think it reads easier if you start a new paragraph here, where you start to present the components.
A: Corrected
617-8: I don't understand what you intended to say here. What do you mean by "for"? Re-write the sentence to clarify.
A: Corrected
620: Change "," to ";".
A: Corrected
623: Remove ".".
A: Corrected
624-5: Yes, but component 2 also seems to separate Ctrl from the others. Is that right
Q: The observation is correct. The sentence is rewritten to make the information clearer.
627-8: Sentence fragment.
A: Corrected
635-6: Sentence fragment.
A: Corrected
639: score plot
A: Corrected
641: Add a "." after "treatments".
A: Corrected
641: score plot
A: Corrected
643: Change "In red show" to "Red indicates
A: Corrected
643: More percentage than what?
A: This sentence refers to the parameters with the highest variance explanation percentages. The sentence is corrected in the text to make the idea more straightforward.
643: How did you set the cutoff for "more percentage"? It doesn't seem to be a certain percentage. Nor does it seem to be a certain number of parameters, like the top 5.
A: We selected the parameters with the highest percentages that explained more than 50% of the data variance per component. For example, for component 1, the cutoff limit for determining the main parameters was up to the “Filo spines FC” variable (Figure 8C), as the sum of the percentages of variance explained by this variable and the preceding ones was 54%.
645: What do you mean by "n =3"? Is that the number of principal components?
A: The value of "n=3" refers to the number of data used per parameter.
647: For sections 2 & 3, the first text in each of those sections is a numbered subsection: 2.1 & 3.1. To be consistent, you may number the first paragraph of the Discussion as 4.1, and then renumber the subsequent subsections.
A: Corrected
648: mechanisms
A: Corrected
651-652: Saying "also" two times makes the conclusions seem redundant and tedious. Re-write to tie your 3 main points together in a more interesting way, for example by noting that your observations go from the big to the small: first at the behavioral level, then at the cellular level, and finally at the molecular level.
A: We corrected as follows:
654-656: In general, introduce data in the Results, then interpret it in the Discussion. Don't introduce data for the first time in the Discussion. Thus, insert this observation into the Results, in an expanded form.
A: We thank this observation. In the new version of our manuscript, we included a subsection in the results section for describing the correlation matrix:
The association among the variables analyzed per group is shown in the correlation matrix constructed for each group to assess the degree and type of association among these phenomena, considering the main parameters analyzed (Figure 10). Supplementary Table 2 shows the r and p values of all significant correlations. In general, it can be observed that there are different patterns of correlation in each of the experimental groups. It may suggest that stress and treatments with rTMS and FLX act by modifying different processes. Among the main significant correlations that were identified, in the Sham group, there exists a negative correlation between the number of immature spines and the total number of dendritic spines (r= -0.997, p= 0.049) and with the mature spines in the FC (r= -1.00, p= 0.014). There is a negative correlation between the DNA demethylation in DG with the total dendritic spines in FC (r= -0.997, p= 0.049) and with the DNA methylation of the mature neurons of DG (r= -1.00, p= 0.020). In the rTMS group, epigenetic repressor mechanisms such as DNA methylation of the DG and histone trimethylation of the FC correlate negatively with the number of immature spines of the FC (r= -0.99, p= 0.002, r= -0.999, p= 0.022 respectively). In addition, there exists a positive correlation between the immunoreactivity of RC3 of the FC with the trimethylation of histone H3 in the DG (r= 0.998, p=0.037). In the case of the FLX group, there is a positive correlation between the acetylation of histone H2B and the complexity of the pyramidal neurons of the FC (r= 0.998, p= 0.037). There is a negative correlation of the trimethylation of histone H3 in the DG with the mature and intermediate dendritic spines in the Hp (r= -0.999, p= 0.030, r= -0.998, p= 0.043, respectively). Overall, these data may suggest that while both rTMS and FLX have similar effects on DS, the molecular mechanisms by which they generate them may imply different modifications at the epigenetic level.
662: "most paradigms"? Do you mean "paradigms most commonly"?
A: Corrected
662-3: "because reproduces"? Do you mean "because it reproduces"?
A: Corrected
665-6: This list is a bit confusing because one might interpret it as meaning a deterioration in 3 things: CS, anhedonia, and despair behaviors. To avoid that possibility, you may change the order to put "deterioration in the CS" last.
A: Corrected
669: Change "C57Bl/6" to "C57BL/6".
A: Corrected
669: Delete "Also, ".
A: Corrected
670: reversed
A: Corrected
672: shown
A: Corrected
672-3: Sentence fragment.
A: Corrected
675: generated
A: Corrected
677: Change "increased in the immobility behavior" to "immobility induced".
A: Corrected
680: Omit the comma. Omit "may".
A: Corrected
681: Change "been" to "with".
A: Corrected
682: induce
A: Corrected
687: probed
A: Corrected
688: [29], [63].
A: Corrected
689: Change "5. Hz" to " 5 Hz".
A: Corrected
697: Why do you say "however"? It seems that this finding from previous studies is consistent with, rather than contradictory to, your findings.
Q: The results are consistent, the handwriting is corrected.
698: decreased
A: Corrected
699-700: Why do you say "In contrast"? That's the same as what you found: that thin type spines are most significantly affected by stress.
A: Previously reported data indicate that in the hippocampus, CUMS affects mainly mature spines (mushroom-like). While in our work, we found that CUMS mainly affects the intermediate spines (thin type). Therefore, we contrast the data and explain this difference because the hippocampal areas analyzed in the previous literature and our work are differ.
700: Change "are" to "were".
A: Corrected
701: cell
A: Corrected
702: Change ", CA1 region" to " or CA1 regions".
A: Corrected
702-3: Add a comma before and after "as pointed out by Qio et al. 2016 [10]".
A: Corrected
706: Add a comma before " confirming".
A: Corrected
709: matrices
A: Corrected
711-2: Sentence fragment.
A: Corrected
714: Change ", shows" to " showed".
A: Corrected
716: generated
A: Corrected
718: Use some sort of punctuation to set off "1.14 Teslas", or precede it by "and ".
A: Corrected
720: That's only a ~35% higher magnetic field. Is that really high intensity? Maybe change "high" to "somewhat higher". In fact, I looked at Table 1 of reference [67] and didn't see that 1.55 T generates the opposite effect. Rather, it looked like both 1.14 T and 1.55 T grew spines (though 1.55 T was less effective than 1.14 T).
A: In the article by Ma et al. (2013) (reference 67), they used intensities of 1.14 and 1.55 Teslas as low and somewhat higher intensity, respectively. Although the difference between the two is not significant, we have revised the text and changed the term “high intensity” to “somewhat higher intensity.” As you mentioned, in the Ma et al. (2013) article, both intensities showed a more significant effect than control groups, although the effect of 1.55 Teslas was lower than 1.14 Teslas. We have made the appropriate corrections in the text.
730: Move the comma after "[69]".
A: Corrected
734-5: "such SYP and RC3"?
A: Corrected
740: Change "synapse dependent of” to “synapses dependent on”.
A: Corrected
769-70: “functions that have also been found to be altered in patients with depression [85].” seems to be stuck on to the sentence without making sense.
- This sentence was included to highlight that patients with depression often experience decreased cognitive abilities, which can be reversed with rTMS. Although our study does not focus on the effect of rTMS on cognition in a model of depressive behaviors, it is mentioned because the frontal cortex and hippocampus play a role in various cognitive processes.
771-2: This section subheading is too complicated.
A: Corrected
773-4: There are two “and”s in this list. Just use “and” before the final item in a list.
A: Corrected
775: “was found to decrease of”?
Q: The sentence is corrected, and it is pointed out that it was a trend.
775-6: This contradicts your Figure 5B, which does not show significance.
A: Although the one-way ANOVA and Bonferroni post hoc test showed no significant differences between the Ctrl and Sham groups, a 21% decrease in acetylation can be observed in the Sham group compared to the Ctrl group. Significant differences were found when performing a t-test between these two groups (p=0.029). Therefore, it is possible that CUMS is decreasing acetylation, and this trend is consistent with existing literature.
778-9: “, H3 [19] and H5 [20] in the hippocampus and nucleus accumbens in models of social defeat stress.” seems to be stuck on to the sentence without making sense.
A: Corrected
783: Use the standard spelling of this mouse line. Search for it online, find the correct spelling, and use that.
A: Corrected
784-5: This contradicts your Figure 5H, where you wrote “Stress and the therapeutic interventions do not modify the H3K9me3 in the DG.”
A: Corrected
787: “C57Bjl6”?
A: Corrected
794: Add a comma after “[87]”.
A: Corrected
797: Add “of “ before “histones”.
A: Corrected
800: Change “vs” to “and”.
A: Corrected
809: Change “GD” to “DG”.
A: Corrected
809-11: I wouldn’t say that your data necessarily contrast with the Reszka study. That’s because you looked at the DG, while Reszka et al looked at blood. But if you want to compare the two studies, you could argue why we may expect that methylation would exhibit similar effects in brain as in blood. However, you just said in the previous paragraph that DNA methylation is particularly abundant in the brain. Presumably, that’s in contrast to the rest of the body.
A: We eliminated the information of the referred study.
811-2: I wrote the above comment before reading this sentence. It’s good you note this. However, it may be best not to even say there’s a contrast in the first place because saying there’s a contrast may mislead and confuse readers and waste their time.
A: Based on the observation of this reviewer, we decided to eliminate the information of the referred study.
826: Add a comma before “ who”.
A: Corrected
832: Change “the dendritic spines” to “dendritic spine”.
A: Corrected
843: Change “GD” to “DG”.
A: Corrected
844-8: There are two sentences stuck together are a grammar error, it’s a comma splice.
A: Corrected
856-7: Change “, in the case of rTMS, they do” to “ rTMS does”.
A: Corrected
859: Change “GD” to “DG”.
A: Corrected
877: facilitates
A: Corrected
897-8: “Its use in the biomedical area (psychiatric, neurodegenerative, motor diseases, etc.) is enhanced and expanded.” Explain. How, when, why? Do you mean by the results in your paper? Or do you mean in general in recent years?
A: This sentence refers to the potential use of rTMS in recent years. The writing is modified to make the idea more straightforward.
906-911: There are a bunch of missing periods on some initials. There’s an extra quotation mark.
A: Corrected
913: ..
A: Corrected
916: Missing period
928: World Health Organization COVID-19 Pandemic... implies that it is the WHO’s pandemic. Add a period after 0the author and before the title.
A: Corrected
939: Add the journal name and volume number.
A: Corrected
954-962: Is reference 14 the same reference as 15?
A: No, they are different items. There may be confusion because they are both by the same author and in the same year but are different articles.
1011-3: Effects. Add the journal name and volume number.
A: Corrected
1014-5: Incomplete list of authors. Wrong journal name. Wrong volume number. Wrong page number.
A: Corrected
1020-4: Reference 39 and 40 refer to the same paper.
A: Corrected
1057-61: References 56 and 57 refer to the same paper.
A: Corrected
1065-8: Remove the letters and numbers after each author’s name. Use initials for authors, just as for other references.
A: Corrected
1069-70: Check the authors’ names.
A: Corrected
1075-7: Check the authors’ names.
A: Corrected
1084: Add the journal name.
A: Corrected
1086: Add the journal name.
A: Corrected
1092: Add the journal name.
A: Corrected
1144: DNA
A: Corrected
1155-6: Is an author actually named C.H. Disease?!
A: Corrected
Supplementary: Change “performed” to “constructed”.
A: Corrected
Supplementary: Change “groups” to “groups,”.
A: Corrected
Supplementary: Change “In general, it can be observed that there are different patterns of correlation in each of the experimental groups,“ to “In general, it can be observed that there are different patterns of correlation in each of the experimental groups;“.
A: Corrected
Supplementary: Change “act modifying” to “act by modifying”.
A: Corrected
Supplementary: Delete “it observed that “.
A: Corrected
Supplementary: Change “spines with” to “spines and”
A: Corrected
Supplementary: Change “p= 0.49” to “p= 0.049”.
A: Corrected
Supplementary: Change “Also exists” to “There is”.
A: Corrected
Supplementary: Change “p= 0.49” to “p= 0.049”.
A: Corrected
Supplementary: Change “Also exists” to “There is”.
A: Corrected
Supplementary: This sentence is complicated, so you may split it into two: “Also exists a negative correlation of the DNA methylation of mature neurons and the trimethylation of histone H3 in the DG with the mature and intermediate dendritic spines in the Hp (r= -0.999, p= 0.030, r= -0.998, p= 0.043 respectively).”
A: Corrected.
In the case of the FLX group, there is a positive correlation between the acetylation of histone H2B and the complexity of the pyramidal neurons of the FC (r= 0.998, p= 0.037). Also, there is a negative correlation between the trimethylation of histone H3 in the DG with the mature DS in the Hp (r= -0.999, p= 0.030). A similar correlation was seen between intermediate dendritic spines in the Hp and the trimethylation of histone H3 in the DG (r= -0.998, p= 0.043).
Bibliography responses.
- Eid, R.S.; Gobinath, A.R.; Galea, L.A.M. Sex Differences in Depression: Insights from Clinical and Preclinical Studies. Prog. Neurobiol. 2019, 176, 86–102, doi: 10.1016/j.pneurobio.2019.01.006.
- Franceschelli, A.; Herchick, S.; Thelen, C.; Papadopoulou-Daifoti, Z.; Pitychoutis, P.M. Sex Differences in the Chronic Mild Stress Model of Depression. Behav. Pharmacol. 2014, 25, 372–383, doi:10.1097/FBP.0000000000000062.
- Oyola MG, Handa RJ. Hypothalamic-pituitary-adrenal and hypothalamic-pituitary-gonadal axes: sex differences in regulation of stress responsivity. Stress. 2017 Sep;20(5):476-494. doi: 10.1080/10253890.2017.1369523. Epub 2017 Aug 31. PMID: 28859530; PMCID: PMC5815295.
- Sterrenburg, L.; Gaszner, B.; Boerrigter, J.; Santbergen, L.; Bramini, M.; Elliott, E.; Chen, A.; Peeters, B.W.M.M.; Roubos, E.W.; Kozicz, T. Chronic Stress Induces Sex-Specific Alterations in Methylation and Expression of Corticotropin-Releasing Factor Gene in the Rat. PLoS One 2011, 6, 1–14, doi: 10.1371/journal.pone.0028128.
- Kushwah, N.; Jain, V.; Deep, S.; Prasad, D.; Singh, S.B.; Khan, N. Neuroprotective Role of Intermittent Hypobaric Hypoxia in Unpredictable Chronic Mild Stress Induced Depression in Rats. PLoS One 2016, 11, 1–20, doi: 10.1371/journal.pone.0149309.
- Ueno, H.; Takahashi, Y.; Murakami, S.; Wani, K.; Matsumoto, Y.; Okamoto, M.; Ishihara, T. Effect of Simultaneous Testing of Two Mice in the Tail Suspension Test and Forced Swim Test. Sci. Rep. 2022, 12, 1–14, doi:10.1038/s41598-022-12986-9.
- De Kloet, E.R.; Molendijk, M.L. Coping with the Forced Swim Stressor: Towards Understanding an Adaptive Mechanism. Neural Plast. 2016, 2016, doi:10.1155/2016/6503162.
- Can, A.; Dao, D.T.; Arad, M.; Terrillion, C.E.; Piantadosi, S.C.; Gould, T.D. The Mouse Forced Swim Test. J. Vis. Exp. 2011, 4–8, doi:10.3791/3638.
- Liu, M.; Yin, C.; Zhu, L.; Zhu, X.; Xu, C.; Luo, C.; Chen, H.; Zhu, D.; Zhou, Q. Sucrose Preference Test for Measurement of Stress-Induced Anhedonia in Mice. Nat. Protoc. 2018, 1–13, doi:10.1038/s41596-018-0011-z.
- Williams, L.J. Newman-Keuls Test and Tukey Test 1 Pairwise Comparisons. Encylopedia Res. Des. 2010, 1, 1–11.
- Shaun, T. Student’s t Table (Free Download) | Guide & Examples. Available online: https://www.scribbr.com/statistics/students-t-table/.
- Schröder, H.; Moser, N.; Huggenberger, S. Neuroanatomy of the Mouse; 2020; ISBN 9783030198978.
We want to thank this reviewer's observations and hope that our answers satisfactorily respond to the questions of this reviewer.

Round 2
Reviewer 1 Report
The authors have appropriately revised the manuscript according to the comments made by the reviewer. I agree with acceptance of the revised manuscript in Cells.
Author Response
We want to thank this reviewer's observations that highlight our study's relevance.
Reviewer 2 Report
I think the manuscript improved notably after revision.
However, I am still not sure about the formula used for calculating sucrose preference index. It seems that authors are using something like this to reach the values presented in figure 3C: SP index %= (sucrose consumption/(total liquid consumed))x100.
Author Response
REVIEWER 2: I think the manuscript improved notably after revision.
Q1. However, I am still not sure about the formula used for calculating sucrose preference index. It seems that authors are using something like this to reach the values presented in figure 3C: SP index %= (sucrose consumption/ (total liquid consumed)) x 100.
A1. We apologize for this mistake. In the revised version (R1), we wrote: "The parameters evaluated in this test were the net sucrose consumption and the sucrose preference index, which takes water consumption into account and is calculated as follows: (Sucrose consumed x (sucrose consumed + natural water consumed)/100))." However, the correct writing is as this reviewer observed: "The parameters evaluated in this test were the net sucrose consumption and the sucrose preference index, which takes water consumption into account and is calculated as follows: (Sucrose consumed / (sucrose consumed + natural water consumed)/100)."
We want to thank this reviewer's observations that highlight our study's relevance.
Reviewer 3 Report
Repetitive transcranial magnetic stimulation (rTMS) is a valuable emerging option for treating depression. It is important to explore mechanisms by which rTMS works. This paper is a worthwhile examination of the effects of rTMS on neuronal structures and epigenetics in mice.
In the original version, there were a huge number of careless mistakes throughout most of the paper, and I tried to list them. The authors have now corrected most of them. I have also made some errors, such as with q-values, which the authors explained. There are still some errors in the paper, and I tried to list them below.
Lines 129-131: There are some language errors in these two sentences. It seems like two sentences were spliced together.
Line 140: The degree symbol should be a superscript.
Lines 158: Written this way, the 4th group seems to have been treated only with FLX. To show that it was also exposed to CUMS, change "FLX" to "mice exposed to CUMS treated with FLX".
Line 188: Change "18 hours. Then" to "18 hours, then".
Lines 195-196: The formula you show for the sucrose preference index seems wrong. Shouldn't it be "sucrose consumed / (sucrose consumed + natural water consumed) x 100"?
Line 241: Change "rate" to "ratio of".
Line 247: Change "Immunohistochemistry" to "immunohistochemistry".
Line 281: Change "was considered" to "were considered".
Figure 3C: To be consistent, change the title from "Preference sucrose index" to "Sucrose preference index".
Figures 3D and 3E: You seem to have switched the titles on these two graphs.
Line 392: Change "Correlation Pearson" to "Pearson Correlation".
Line 399: Delete "In panel F (**), symbols represent a significative correlation (* p ≤ 0.005)." and the asterisks from Figure 3F because they don't add any information. Move "N= 37." to the caption for Figure 3F (line 392).
Lines 446-7: I had originally understood this sentence to mean that the CUMS decreased filo spines, and that it decreased thin spines, and that it decreased mushroom spines. That's why I estimated the values for one of the spine types (filo) and did a t-test, which showed that filo spines appeared not to be significantly changed. But after your explanation, I now understand that you combined all types of spines and found a significant difference between control and CUMS. Figure 4F already showed that, so I don't see the need to repeat the finding of 4F by doing the same analysis in 4G. If you are going to go to the effort to classify types of spines in Figure 4G, why then lump all types together again to analyze them? It would make more sense to analyze each type separately and report the effects. If the effects are not statistically significant, then report that. If you then want to lump all types of spines together, you can do that and report it afterward. But the main point of 4G is to analyze each type of spine separately. Otherwise, omit 4G.
Line 495: Change " panels, F, and I (*)," to " panels F and I, (*)".
Line 496: Change "panels, G and J (&) symbol" to "panels G and J, (&) symbols".
Line 498: Change "G. (*) symbol represent" to "G, (*) symbols represent". Change "DS DS" to "DS".
Line 499: Change " J. (*) symbol represents the significative differences between groups consider the DS" to " J, (*) symbols represent significant differences between groups considering the".
Line 510: If this is a new paragraph, indent it.
Figure 5B, 5D, 5F, & 5H: Since the units are arbitrary, you can divide them by 1000 so that you can simplify them by removing the factor of 1000 in the labels. Same for Figure 6, 7, and 8.
Line 594: Figure 8 shows 5hmC, not Figure 7.
Line 596: Indent the new paragraph.
Line 633: Move the comma from after "and" to before "and".
Line 647-8: Instead of saying "concerning the CUMS group", you might convey your message clearer if you said "compared to Sham", or something like that. The p values you then show are not the relevant p values. You should show the p values for comparing rTMS vs. Sham and for comparing FLX vs. Sham.
Line 656: Change "citosin" to "cytosine".
Line 657: Change "5-hidroxy-methyl-citosin" to "5-hydroxy-methyl-cytosine".
Line 661: You don't need to keep spelling out 5-hydroxy-methyl-cytosine after the first time. But if you do, change "5-hidroxy-methyl-citosin" to "5-hydroxy-methyl-cytosine".
Figure 9 and 10: Change "5mC (ng) DG" to "5mC (ng/µl) DG".
700 and 702: Change "score plots" to "score plot".
702: Add a "." after "treatments".
704-5: What do you mean by "a higher percentage"? You gave an explanation to me, but you should explain clearly to the readers in the paper.
Figure 10: I'm sorry that I didn't notice this in the first review, but I now realize that it is remarkable that many of the correlations are very close to perfect (r~1 or -1). Is that because n is only 3? Is it even meaningful to examine correlations for such an extremely small number of points? Two points will always show perfect correlation, and you're only adding one point to that perfect correlation. This deserves comment and explanation. Also, you correlated many, many pairs of variables; how did you correct statistical significance for multiple comparisons? Because of these problems, I suggest deleting all this analysis of correlations (section 3.8, Figure 10, Supplementary Table 2, and elsewhere in the paper). I realize that animal experiments are expensive and time-consuming, but n=3 is often too few to draw more than very simple conclusions.
776: Omit the comma (or add a comma after "results").
871: tends
993: Change "rTMScan" to "rTMS".
1012: There is no period at the end of the sentence
1024: For reference 2, add a period after the author and before the title.
1034-5: For reference 7, add the journal name and volume number.
1050-5: Is reference 14 the same reference as 15?
1109-11: For reference 35, add the journal name.
1112-3: For reference 36, there are several things wrong. Incomplete list of authors. Wrong journal name. Wrong volume number. Wrong page number.
1161-3: References 39 and 58 refer to the same paper.
1248: For reference 91, change "Dna" to "DNA".
1272-3: For reference 101, the journal name is Prog Mol Biol Transl Sci.
For all the references, there were so many errors that you should double check all the references for duplicates and other errors.
Supplementary Table 1: Change "Week acclimatization" to "Acclimatization week".
Supplementary Table 1: Change "Preference sucrose test" to "Sucrose preference test".
See above Comments and Suggestions for Authors
Author Response
REVIEWER 3: Repetitive transcranial magnetic stimulation (rTMS) is a valuable emerging option for treating depression. It is important to explore mechanisms by which rTMS works. This paper is a worthwhile examination of the effects of rTMS on neuronal structures and epigenetics in mice.
In the original version, there were a huge number of careless mistakes throughout most of the paper, and I tried to list them. The authors have now corrected most of them. I have also made some errors, such as with q-values, which the authors explained. There are still some errors in the paper, and I tried to list them below.
Lines 129-131: There are some language errors in these two sentences. It seems like two sentences were spliced together.
A: Sentences are rewritten to make them more understandable in the new version of the text,
“A MagPro R30 (MagVenture, Georgia, USA) stimulator with a coil adapted for rodents (Cool-40) was used.”
Line 140: The degree symbol should be a superscript.
A: Corrected.
Lines 158: Written this way, the 4th group seems to have been treated only with FLX. To show that it was also exposed to CUMS, change "FLX" to "mice exposed to CUMS treated with FLX".
A: Corrected.
Line 188: Change "18 hours. Then" to "18 hours, then".
A: Corrected.
Lines 195-196: The formula you show for the sucrose preference index seems wrong. Shouldn't it be "sucrose consumed / (sucrose consumed + natural water consumed) x 100"?
A: We apologize for this mistake. In the revised version (R1), we wrote: "The parameters evaluated in this test were the net sucrose consumption and the sucrose preference index, which takes water consumption into account and is calculated as follows: (Sucrose consumed x (sucrose consumed + natural water consumed)/100))." However, the correct writing is as this reviewer observed: "The parameters evaluated in this test were the net sucrose consumption and the sucrose preference index, which takes water consumption into account and is calculated as follows: (Sucrose consumed / (sucrose consumed + natural water consumed)/100)."
Line 241: Change "rate" to "ratio of".
A: Corrected.
Line 247: Change "Immunohistochemistry" to "immunohistochemistry".
A: Corrected.
Line 281: Change "was considered" to "were considered".
A: Corrected.
Figure 3C: To be consistent, change the title from "Preference sucrose index" to "Sucrose preference index".
A: Corrected.
Figures 3D and 3E: You seem to have switched the titles on these two graphs.
A: Corrected.
Line 392: Change "Correlation Pearson" to "Pearson Correlation".
A: Corrected.
Line 399: Delete "In panel F (*), symbols represent a significative correlation (p ≤ 0.005)." and the asterisks from Figure 3F because they don't add any information. Move "N= 37." to the caption for Figure 3F (line 392).
A: Corrected.
Lines 446-7: I had originally understood this sentence to mean that the CUMS decreased filo spines, and that it decreased thin spines, and that it decreased mushroom spines. That's why I estimated the values for one of the spine types (filo) and did a t-test, which showed that filo spines appeared not to be significantly changed. But after your explanation, I now understand that you combined all types of spines and found a significant difference between control and CUMS. Figure 4F already showed that, so I don't see the need to repeat the finding of 4F by doing the same analysis in 4G. If you are going to go to the effort to classify types of spines in Figure 4G, why then lump all types together again to analyze them? It would make more sense to analyze each type separately and report the effects. If the effects are not statistically significant, then report that. If you then want to lump all types of spines together, you can do that and report it afterward. But the main point of 4G is to analyze each type of spine separately. Otherwise, omit 4G.
A: A: We understand the point of this reviewer. We consider it important to maintain panel G in Figure 4 because a morphological analysis of the DS was carried out to determine treatments' impact on any particular type of DS. Thus, we analyzed the results per spine type using a two-way ANOVA considering factors of treatment and spine type. To clarify this point, in the new version of our manuscript, we rewritten the text as follows:
Therefore, we analyzed the effect of treatments on DS types based on their morphology in the FC and DG (Figure 4G, J respectively). In the FC (Figure 4G), we did not find an interaction between the factors (Figure 4G; Two-way ANOVA. Treatment X DS type: F6,47 = 2, p = 0.074). However, the main effect of treatment (F3,47 = 10, p < 0.001) confirmed that the CUMS decreased the number of all types of DS, regardless of type, compared to the control group (p < 0.001). This effect was reversed by 5 Hz rTMS (p = 0.013), but FLX tended to reverse the effect of chronic stress (p = 0.082). Also, the main effect of DS type (F2,47 = 88, p < 0.001) showed that in all groups, mushroom-like spines were the most abundant compared with filopodia (p < 0.001) and thin (p < 0.001) DS type (Figure 4G).
Line 495: Change " panels, F, and I ()," to " panels F and I, ()".
A: Corrected.
Line 496: Change "panels, G and J (&) symbol" to "panels G and J, (&) symbols".
A: Corrected.
Line 498: Change "G. () symbol represent" to "G, () symbols represent". Change "DS DS" to "DS".
A: Corrected.
Line 499: Change " J. () symbol represents the significative differences between groups consider the DS" to " J, () symbols represent significant differences between groups considering the".
A: Corrected.
Line 510: If this is a new paragraph, indent it.
A: Corrected.
Figure 5B, 5D, 5F, & 5H: Since the units are arbitrary, you can divide them by 1000 so that you can simplify them by removing the factor of 1000 in the labels. Same for Figure 6, 7, and 8.
A: Corrected.
Line 594: Figure 8 shows 5hmC, not Figure 7.
A: Corrected.
Line 596: Indent the new paragraph.
A: Corrected.
Line 633: Move the comma from after "and" to before "and".
A: Corrected.
Line 647-8: Instead of saying "concerning the CUMS group", you might convey your message clearer if you said "compared to Sham", or something like that. The p values you then show are not the relevant p values. You should show the p values for comparing rTMS vs. Sham and for comparing FLX vs. Sham.
A: The sentence is rewritten to make the message more understandable:
“Mice treated with rTMS or FLX did not show significant changes compared to Sham group (p = 1.00)”
Line 656: Change "citosin" to "cytosine".
A: Corrected.
Line 657: Change "5-hidroxy-methyl-citosin" to "5-hydroxy-methyl-cytosine".
A: Corrected.
Line 661: You don't need to keep spelling out 5-hydroxy-methyl-cytosine after the first time. But if you do, change "5-hidroxy-methyl-citosin" to "5-hydroxy-methyl-cytosine".
A: Corrected.
Figure 9 and 10: Change "5mC (ng) DG" to "5mC (ng/µl) DG".
A: Corrected.
700 and 702: Change "score plots" to "score plot".
A: Corrected.
702: Add a "." after "treatments".
A: Corrected.
704-5: What do you mean by "a higher percentage"? You gave an explanation to me, but you should explain clearly to the readers in the paper.
A: The observation is taken into account and the method for choosing the main parameters that explain the variance for each component is added in the new version:
“Red indicates the main parameters with a higher percentage that explain the variance by component. The cut-off points for choosing these parameters were that the sum of all of them should explain 50% of the variance.”
Figure 10: I'm sorry that I didn't notice this in the first review, but I now realize that it is remarkable that many of the correlations are very close to perfect (r~1 or -1). Is that because n is only 3? Is it even meaningful to examine correlations for such an extremely small number of points? Two points will always show perfect correlation, and you're only adding one point to that perfect correlation. This deserves comment and explanation. Also, you correlated many, many pairs of variables; how did you correct statistical significance for multiple comparisons? Because of these problems, I suggest deleting all this analysis of correlations (section 3.8, Figure 10, Supplementary Table 2, and elsewhere in the paper). I realize that animal experiments are expensive and time-consuming, but n=3 is often too few to draw more than very simple conclusions.
A: Your observation is very valuable. We agree with the point to eliminate the correlation sections.
776: Omit the comma (or add a comma after "results").
A: Corrected.
871: tends
A: Corrected.
993: Change "rTMScan" to "rTMS".
A: Corrected.
1012: There is no period at the end of the sentence
A: Corrected.
1024: For reference 2, add a period after the author and before the title.
A: Corrected.
1034-5: For reference 7, add the journal name and volume number.
A: Corrected.
1050-5: Is reference 14 the same reference as 15?
A: Your observation is correct. The reference in the text is corrected:
“Smrt, R.D.; Zhao, X. Epigenetic regulation of neuronal dendrite and dendritic spine development. Front Biol. 2010, 4, 304-323. doi: 10.1007/s11515-010-0650-0.
1109-11: For reference 35, add the journal name.
A: Corrected.
1112-3: For reference 36, there are several things wrong. Incomplete list of authors. Wrong journal name. Wrong volume number. Wrong page number.
A: Corrected.
1161-3: References 39 and 58 refer to the same paper.
A: We apologize for the confusion. The reference is corrected in the new version of the text.
1248: For reference 91, change "Dna" to "DNA".
A: Corrected.
1272-3: For reference 101, the journal name is Prog Mol Biol Transl Sci.
A: Corrected.
For all the references, there were so many errors that you should double check all the references for duplicates and other errors.
A: Corrected.
Supplementary Table 1: Change "Week acclimatization" to "Acclimatization week".
A: Corrected.
Supplementary Table 1: Change "Sucrose preference test" to "Sucrose preference test".
A: Corrected.
We want to thank the observations of this reviewer, and we hope that our answers satisfactorily respond to the questions of this reviewer.